# C-Planning: An Automatic Curriculum for Learning Goal-Reaching Tasks

**Tianjun Zhang**[*]
UC Berkeley
`tianjunz@berkeley.edu`

**Benjamin Eysenbach**[*]
Carnegie Mellon University
`beysenba@cs.cmu.edu`

**Ruslan Salakhutdinov**
Carnegie Mellon University

**Sergey Levine**
UC Berkeley

**Joseph E. Gonzalez**
UC Berkeley

## Abstract

Goal-conditioned reinforcement learning (RL) can solve tasks in a wide range of domains, including navigation and manipulation, but learning to reach distant goals remains a central challenge to the field. Learning to reach such goals is particularly hard without any offline data, expert demonstrations, and reward shaping. In this paper, we propose an algorithm to solve the distant goal-reaching task by using planning at training time to automatically generate a curriculum of intermediate states. Our algorithm, Classifier-Planning (C-Planning), frames the learning of the goal-conditioned policies as expectation maximization: the E-step corresponds to planning a sequence of waypoints using graph planning, while the M-step aims to learn a goal-conditioned policy to reach those waypoints. Unlike prior methods that combine goal-conditioned RL with graph search, ours performs planning only during training and not testing, significantly decreasing the compute costs of deploying the learned policy. Empirically, we demonstrate that our method is more sample efficient that prior methods. Moreover, it is able to solve very long horizons manipulation and navigation tasks, tasks that prior goal-conditioned methods and methods based on graph search fail to solve.

## 1 Introduction

Whereas typical RL methods maximize the accumulated reward, goal-conditioned RL methods learn to reach any goal. Arguably, many tasks are more easily defined as goal-reaching problems than as reward maximizing problems. While many goal-conditioned RL algorithms learn policies that can reach nearby goals, learning to reach distant goals remains a challenging problem. Some prior methods approach this problem by performing search or optimization over subgoals at test time. However, these test-time planning methods either rely on graph search (Eysenbach et al., 2019; Savinov et al., 2018), which scales poorly with dimensionality (Hsu et al., 2006), or continuous optimization over subgoals (Nasiriany et al., 2019), which is expensive and can result in model exploitation.

In this paper, we take a different tack and instead use planning at training time to automatically generate a curriculum. When training the agent to reach one goal, our method first determines some intermediate waypoints enroute to that goal. Then, it commands the agent to reach those waypoints before navigating to the final destination. Collecting data in this manner improves the quality of data, allowing the agent to learn to reach distant goals. Importantly, our method does not perform planning at test time, decreasing the computational demands for deploying the learned agent.

Our curriculum does not require manual engineering or prior knowledge of the tasks. Rather, the curriculum emerges automatically when we use expectation maximization (EM) to maximize a lower bound on the probability of reaching the goal. The M-step corresponds to a prior method for goal-conditioned RL, while the E-step corresponds to graph planning. Thus, EM presents a marriage between previous goal-conditioned RL algorithms and graph-planning methods.

---

[*]Equal contribution.



Figure 1: **C-Planning** is an algorithm for goal-conditioned RL that uses an automatic curriculum of waypoints to learn policies that can solve complex tasks. In this manipulation task, the goal requires moving the green puck to the green dot and the red puck to the red dot. Our method learns to solve this task, manipulating multiple objects in sequence, without requiring any reward functions, manual distance functions, or human demonstrations.

The main contribution of this work is a goal-conditioned RL algorithm, C-Planning, that excels at reaching distant goals. Our method uses an automatic curriculum of subgoals to accelerate training. The automatic waypoint sampling ends up generating a curriculum of waypoints, points that are reachable by the current policy but also lead to the goal. We show that our curriculum emerges from applying variational inference to the goal-reaching problem. Unlike prior work, our method does not require graph search or optimization at test-time. Empirically, C-Planning not only matches but surpasses the performance of these prior search-based methods, suggesting that it is not just amortizing the cost of graph search. We empirically evaluate C-Planning on temporally-extended 2D navigation tasks and complex 18D robotic manipulation tasks. C-Planning improves the sample efficiency of prior goal-conditioned RL algorithms and manages to solve more difficult manipulations tasks such as rearranging multiple objects in sequence (see Fig. 1.). To the best of our knowledge, no prior method has learned tasks of such difficulty without requiring additional assumptions.

## 2 RELATED WORK

The problem of learning to achieve goals has a long history, both in the control community Lyapunov (1992) and the RL community (Kaelbling, 1993). Many prior papers approach goal-conditioned RL as a reward-driven, multi-task learning problem, assuming access to a goal-conditioned reward function. One unique feature of goal-conditioned RL, compared to other multi-task RL settings, is that it can also be approached using reward-free methods, such as goal-conditioned behavior cloning (Ding et al., 2019; Gupta et al., 2020; Lynch et al., 2020; Savinov et al., 2018) and RL methods that employ hindsight relabeling (Eysenbach et al., 2020; Kaelbling, 1993; Lin et al., 2019; Schroecker & Isbell, 2020). While goal-conditioned behavior cloning methods are simple to implement and have shown excellent results on a number of real-world settings (Meng et al., 2020; Shah et al., 2020), they are not guaranteed to recover the optimal policy without additional assumptions (e.g., determinism, online data collection). While both methods excel at certain control tasks, both often struggle to solve tasks with longer horizons.

To solve longer-horizon tasks, prior work has combined goal-conditioned RL with graph search, noting that goal-conditioned value functions can be interpreted as dynamical distances (Eysenbach et al., 2018; Huang et al., 2019; Nasiriany et al., 2019; Savinov et al., 2018). These prior methods typically proceed in two phases: the first phase learns a goal-conditioned policy, and the second phase combines that policy with graph search. While these methods have demonstrated excellent results on a number of challenging control tasks, including real-world robotic navigation (Meng et al., 2020; Shah et al., 2020), the stage-wise approach has a few limitations. First, the post-hoc use of graph search means that graph search cannot improve the underlying goal-conditioned policy. It is well known that the performance of RL algorithms is highly dependent on the quality of the collected data (Kakade & Langford, 2002; Kumar et al., 2020; Levine et al., 2020). By integrating planning into the learning of the underlying goal-conditioned policy, our method will improve the quality of the data used to train that policy. A second limitation of these prior methods is the cost of deployment: choosing actions using graph search requires at least $\mathcal{O}(|\mathcal{V}|)$ queries to a neural network.[1] In contrast, our method performs planning at training time rather than test time, so the cost of deployment is $\mathcal{O}(1)$. While this design decision does increase the computational complexity of training, it significantly decreases the latency at deployment. Our method is similar to RIS (Chane-Sane et al., 2021), which also performs planning during training instead of deployment. Our method differs from RIS in how planning is used: whereas RIS modify the *objective*, our method

---

[1]While computing all pairwise distances requires $\mathcal{O}(|\mathcal{V}|^2)$ time, only $|\mathcal{V}|$ edges change at each time step. The cost of computing the remaining edges can be amortized across time.

uses planning to modify the *data*. This difference not only unifies the objectives for RL and graph planning, but also significantly improves the performance of the algorithm. , a subtle distinction that avoids favoring the learned policy and unifies the objectives for RL and graph planning. We demonstrate the importance of this difference in our experiments.

Effectively solving goal-conditioned RL problems requires performing good exploration. In the goal-conditioned setting, the quality of exploration depends on how goals are sampled, a problem studied in many prior methods (Eysenbach et al., 2018; Florensa et al., 2017; 2018; Pitis et al., 2020; Pong et al., 2020; Ren et al., 2019; Zhang et al., 2020; 2021). These methods craft objectives that try to optimize for learning progress, and the resulting algorithms achieve good results across a range of environments. Our method differs from these methods in that the method for optimizing the goal-conditioned policy and the method for sampling waypoints are jointly optimized using the same objective. We demonstrate the importance of this difference in our experiments.

Our approaches can also be viewed as one Hierarchical RL (Dietterich, 2000; Levy et al., 2017; Vezhnevets et al., 2017) method with the distance function specified by the classifier. However, one important difference between C-Planning and many other HRL methods is that our approach doesn't require training a high-level policy for generating subgoals for the lower-level policy to reach (Gupta et al., 2020; Levy et al., 2017; Nachum et al., 2018). Comparing with approaches try to build a graph/tree for planning (Eysenbach et al., 2019; Huang et al., 2019; Nasiriany et al., 2019), our approach doesn't require any planning at deployment, saves a lot of computation time.

Our method builds on the idea that reinforcement learning can be cast as inference problems (Attias, 2003; Levine, 2018; Rawlik et al., 2013; Theodorou et al., 2010; Todorov, 2007; Ziebart, 2010). The observation that variational inference for certain problems corresponds to graph planning is closely related to Attias (2003). Our work extends this inferential perspective to hierarchical models.

## 3 PRELIMINARIES

We introduce the goal-conditioned RL problem and a recent goal-conditioned RL algorithm, C-learning, upon which our method builds.

**Goal-Conditioned RL.** We focus on controlled Markov processes defined by states $s_t \in \mathcal{S}$ and actions $a_t$. The initial state is sampled $s_0 \sim p_0(s_0)$ and subsequent states are sampled $s_{t+1} \sim p(s_{t+1} \mid s_t, a_t)$. Our aim is to navigate to the goal states, $s_g \in \mathcal{S}$, which are sampled from $s_g \sim p_g(s_g)$. We define a policy $\pi(a_t \mid s_t, s_g)$ conditioned on both the current state and the goal. The objective is to maximize the probability (density) of reaching the goal in the future.

To derive our method, or any other goal-conditioned RL algorithm, we must make a modeling assumption about *when* we would like the agent to reach the goal. This modeling assumption is only used to derive the algorithm, not for evaluation. Formally, let $\Delta \in \mathbb{N}^+$ be a random integer indicating when we reach the goal. The user specifies a prior $p(\Delta)$. Most prior work (implicitly) uses the geometric distribution, $p(\Delta) = \text{GEOM}(1 - \gamma)$. The geometric distribution is ubiquitous because it is easy to incorporate into temporal difference learning. Despite its wide usage, it may not reflect the prior belief on when the agent will solve the task or when the episode will terminate. Thus, while it is natural to use geometric distribution, other choices of distributions can also be consdiered. Prior work has also considered non-geometric distributions (Fedus et al., 2019). In this paper, we use the negative binomial distribution, $p(\Delta) = \text{NEGBINOM}(p = 1 - \gamma, n = 2)$. Given a prior $p(\Delta)$, we define the distribution over future states as

$$p_{p(\Delta)}^{\pi(\cdot \mid \cdot, s_g)}(s_{t+} = s_{t+1} \mid s_t, a_t) = \mathbb{E}_{p(\Delta)} \left[ p_{\Delta}^{\pi(\cdot \mid \cdot, s_g)}(s_{t+\Delta} = s_{t+1} \mid s_t, a_t) \right], \tag{1}$$

where $p_{\Delta}^{\pi}(s_{t+\Delta} \mid s_t, a_t)$ is the probability density of reaching state $s_{t+\Delta}$ exactly $\Delta$ steps in the future when sampling actions from $\pi(a_t \mid s_t, s_g)$. For example, the $\gamma$-discounted state occupancy measure (Ho & Ermon, 2016; Nachum et al., 2019) can be written as $p_{\text{GEOM}(1-\gamma)}^{\pi(\cdot \mid \cdot, s_g)}$. Our objective for goal-reaching is to maximize the probability of reaching the desired goal:

$$\max_{\pi} \mathbb{E}_{p_g(s_g)} \left[ p_{\text{NEGBINOM}(p=1-\gamma, n=2)}^{\pi(\cdot \mid \cdot, s_g)}(s_{t+} = s_g) \right]. \tag{2}$$

**C-Learning.** Our method builds upon a prior method for goal-conditioned RL, C-Learning (Eysenbach et al., 2020). C-Learning learns a classifier for predicting whether a state $s_t$ comes from a

future state density $p_{p(\Delta)}^{\pi(\cdot|\cdot,s_g)}(s_{t+} = s_{t+} \mid s_t, a_t)$ or a marginal state density:

$$p_{p(\Delta)}^{\pi(\cdot|\cdot,s_g)}(s_{t+} = s_{t+}) = \int p_{p(\Delta)}^{\pi(\cdot|\cdot,s_g)}(s_{t+} = s_{t+} \mid s_t, a_t)\, p(s_t, a_t)\, ds_t\, da_t.$$

The Bayes-optimal classifier can be written in terms of these two distributions:

$$C_\theta(s, s_g) = \frac{p_{\text{GEOM}}^{\pi(\cdot|\cdot,s_g)}(s_g \mid s)}{p_{\text{GEOM}}^{\pi(\cdot|\cdot,s_g)}(s_g \mid s) + p_{\text{GEOM}}^{\pi(\cdot|\cdot,s_g)}(s_g)}. \tag{3}$$

In C-Learning, this classifier acts as a value function for training the policy. Our method will use this classifier not only to update the policy, but also to sample waypoints.

## 4 C-PLANNING: GOAL-CONDITIONED RL WITH PLANNING

We now present our method, C-Planning. The main idea behind our method is to decompose the objective of Eq. 2 into a sequence of easier goal-reaching problems. Intuitively, if we want a sub-optimal agent to navigate from some initial state $s_0$ to a distant goal state $s_g$, having the agent try to reach that state again and again will likely prove futile.

Instead, we can command a sequence of goals, like a trail of breadcrumbs leading to the goal $s_g$. To select these waypoints, we will command the policy to reach those states it would visit *if* it were successful at reaching $s_g$. Sampling subgoals in this way can ease the difficulty of training the agent.

At a high-level algorithm summary, our method consists of two steps: using wayping sampling to collect experience and performing standard goal-conditioned RL using the collected experience. The main challenge is the first step, as modeling the distribution over waypoints is difficult. In Sec. 4.1, we solve analytically for waypoint distribution, and then propose a simple practical method to sample from this distribution in Sec. 4.2.

### 4.1 PLANNING AND VARIATIONAL INFERENCE

To formally derive our method for waypoint sampling, we cast the problem of goal-reaching as a latent variable problem. We assume that the agent starts at state $s_0$ and ends at state $s_g$, but the intermediate states that the agent will visit are unknown latent variables. This problem resembles standard latent variable modeling problems (e.g., VAE), where the intermediate state serves the role of the inferred representation. We can thus derive an evidence lower bound on the likelihood of reaching the desired goal:

**Lemma 1.** *The objective $\mathcal{L}$, defined below, is a lower bound on the goal-reaching objective (Eq. 2):*

$$\log p_{\text{NEGBINOM}}^{\pi(\cdot|\cdot,s_g)}(s_{t+} = s_g \mid s_0) \geq \mathbb{E}_{q(s_w|s_g,s_0)}\Big[ \log p_{\text{GEOM}}^{\pi(\cdot|\cdot,s_g)}(s_{t+} = s_g \mid s_w) + \log p_{\text{GEOM}}^{\pi(\cdot|\cdot,s_g)}(s_w \mid s_0)$$
$$- \log q(s_w \mid s_g, s_0)\Big] \triangleq \mathcal{L}(\pi, q(s_w \mid s_g, s_0)).$$

See Appendix A.1 for the proof. The lower bound, $\mathcal{L}$, depends on two quantities: the goal-conditioned policy and an *inferred* distribution over waypoints, $q(s_w \mid s_0, s_g)$. This bound holds for any choice of $q(s_w \mid s_0, s_g)$, and we can optimize this lower bound with respect to this waypoint distribution. We can see more intuition behind the negative binomial distribution here. With the negative binomial distribution, we can decompose a hard problem into a sequence of easier RL problems. Each of these easier (goal-conditioned) RL problems uses geometric distributions, and the sum of geometric random variables is a negative binomial distribution.

We will perform expectation maximization (Dempster et al., 1977) to optimize the lower bound, alternating between estimating waypoint distribution and optimizing the goal-conditioned policy using this waypoint distribution. We now describe these two steps in detail.

**E-Step.** The E-step estimates the waypoint distribution, $q(s_w \mid s_0, s_g)$. The three terms in the lower bound indicate that the sampled waypoint should be reachable from the initial state, the goal state should be reachable from the sampled waypoint, and that the waypoint distribution should have

high entropy. These first two terms resemble shortest-path planning, where distances are measured using log probabilities. Importantly, reachability is defined in terms of the capabilities of the current policy. We can analytically solve for the waypoint distribution:

**Lemma 2.** *The waypoint distribution that optimizes our lower bound (Lemma 1) satisfies:*

$$q^*(\boldsymbol{s}_w \mid \boldsymbol{s}_g, \boldsymbol{s}_0) = \frac{p_{\text{GEOM}}^{\pi(\cdot \mid \cdot, \boldsymbol{s}_g)}(\boldsymbol{s}_g \mid \boldsymbol{s}_w) p_{\text{GEOM}}^{\pi(\cdot \mid \cdot, \boldsymbol{s}_w)}(\boldsymbol{s}_w \mid \boldsymbol{s}_0)}{\int p_{\text{GEOM}}^{\pi(\cdot \mid \cdot, \boldsymbol{s}_g)}(\boldsymbol{s}_g \mid \boldsymbol{s}_w') p_{\text{GEOM}}^{\pi(\cdot \mid \cdot, \boldsymbol{s}_w)}(\boldsymbol{s}_w' \mid \boldsymbol{s}_0) d\boldsymbol{s}_w'}.$$

See Appendix A.1 for a full derivation. In general, accurately estimating the distribution, $q^*$, is challenging. However, in the next section, we develop a simple algorithm that only requires learning a classifier instead of a generative model.

**M-step.** The M-step optimizes the lower bound with respect to the goal-conditioned policy. We can ignore the $-\log q(\boldsymbol{s}_w \mid \boldsymbol{s}_0, \boldsymbol{s}_g)$ term, which does not depend on the policy. The remaining two terms look like goal-conditioned RL objectives, with a subtle but important difference in how the trajectories are sampled. When collecting data, a standard goal-conditioned RL algorithm initially samples a goal and then collects a trajectory where the policy attempts to reach that goal. Our method collects data in a different manner. The agent samples a goal, *then samples an intermediate waypoint that should lead to that goal.* The agent attempts to reach the waypoint *before* attempting to reach the goal. After the trajectory has been collected, the policy updates are the same as a standard goal-conditioned RL algorithm.

**Two mental models.** The algorithm sketch we have presented, which will be fleshed out in the subsequent section, can be interpreted in two ways. One interpretation is that the method performs a soft version of graph planning during exploration, using a distance function of $d(\boldsymbol{s}_1, \boldsymbol{s}_2) = \log p_{\text{GEOM}}^{\pi(\cdot \mid \cdot, \boldsymbol{s}_2)}(\boldsymbol{s}_{t+} = \boldsymbol{s}_2 \mid \boldsymbol{s}_1)$. From this perspective, the method is similar to prior work that performs graph search during test-time (Eysenbach et al., 2019; Savinov et al., 2018). However, our method will perform planning at training time. Extensive ablations in Sec. 5 show that our method outperforms alternative approaches that only perform search during test-time.

The second interpretation focuses on the task of reaching the goal from different initial states. The choice of waypoint distribution, and the policy's success at reaching those waypoints, determines the initial state distribution for this task. Intuitively, the best initial state distribution is one that sometimes starts the agent at the true initial state $\boldsymbol{s}_0$, sometimes starts the agent at the final state $\boldsymbol{s}_g$, and sometimes starts the agent in between. In fact, prior work has formally shown that the optimal initial state distribution is the marginal state distribution of the optimal policy (Kakade & Langford, 2002). Under somewhat strong assumptions,[2] this is precisely what our waypoint sampling achieves. We refer the reader for the discussion and experiments in Appendix. A.

## 4.2 A PRACTICAL IMPLEMENTATION

We describe how we implement the policy updates (M-step) and waypoint sampling (E-step). For policy updates, we simply apply C-learning (Eysenbach et al., 2020). The main challenge is waypoint sampling, which requires sampling a potentially high-dimensional waypoint distribution. While prior work (Florensa et al., 2018) has approached such problems by fitting high-capacity generative models, our approach avoids such generative models and instead use importance sampling.

Let $b(\cdot)$ be the background distribution taking to be the replay buffer. Our algorithm corresponds to the following two-step procedure. First, we sample a batch of waypoints from the buffer. Then, we sample one waypoint from within that batch using the normalized importance weights, $\frac{q(\boldsymbol{s}_w \mid \boldsymbol{s}_g, \boldsymbol{s}_0)}{b(\boldsymbol{s}_w)}$. We can estimate these importance weights using the classifier learned by C-learning (Eq. 3):

**Lemma 3.** *We can write the importance weights in terms of this value function*

$$\frac{q(\boldsymbol{s}_w \mid \boldsymbol{s}_g, \boldsymbol{s}_0)}{b(\boldsymbol{s}_w)} = \frac{C_\theta(\boldsymbol{s}_w, \boldsymbol{s}_g)}{1 - C_\theta(\boldsymbol{s}_w, \boldsymbol{s}_g)} \frac{C_\theta(\boldsymbol{s}_0, \boldsymbol{s}_w)}{1 - C_\theta(\boldsymbol{s}_0, \boldsymbol{s}_w)} Z(\boldsymbol{s}_0, \boldsymbol{s}_g), \tag{4}$$

*where $Z(\boldsymbol{s}_0, \boldsymbol{s}_g)$ is a normalizing constant.*

---

[2] The assumptions are that *(1)* the policy always reaches the commanded waypoint, and that *(2)* that the goal-reaching probabilities $p_{\text{GEOM}}^{\pi(\cdot \mid \cdot, \cdot)}$ reflect the probability that the *optimal* policy reach some state.

**Algorithm 1 C-Planning** performs planning in data collection, modifies C-learning by $L5 \to L6 - 7$. The update for the policy and classifier ($L9$) is the same.

1: $\mathcal{D} \leftarrow \emptyset, N_g, \epsilon_d$
2: **for** $0 \le i \le N$ **do**
3:     Set $n_g \leftarrow 0$ if new episode
4:     $\boldsymbol{s}_0 \sim p_0(\boldsymbol{s}_0), \boldsymbol{s}_g \sim p_g(\boldsymbol{s}_g)$
5:     $\color{red}{\tau \sim p^\pi(\tau \mid \boldsymbol{s}_0, \boldsymbol{s}_g)}$
6:     $\color{green}{\boldsymbol{s}_w, n_g \leftarrow \text{CPLANNING}(n_g, C, s_g)}$
7:     $\color{green}{\tau \leftarrow (\tau_1 \sim p^\pi(\tau \mid \boldsymbol{s}_0, \boldsymbol{s}_w),}$
               $\color{green}{\tau_2 \sim p^\pi(\tau \mid \boldsymbol{s}_w, \boldsymbol{s}_g))}$
8:     $\mathcal{D} \leftarrow \mathcal{D} \cup \{\tau\}$
9:     $\pi, C \leftarrow \text{CLEARNING}(\mathcal{D}, \pi, C)$

**Algorithm 2 C-Planning** samples the intermediate waypoints, then command the agent to reach them.

1: $n_g$: number of waypoints agent has reached in an episode
2: **function** CPLANNING$(n_g, C, s_g)$
3:     **if** $(t = 0$ or $d(\boldsymbol{s}_t, \boldsymbol{s}_w) \le \epsilon_d)$ and $n_g \le N_g$ **then**
4:         $n_g \leftarrow n_g + 1$
5:         Sample M waypoints $\boldsymbol{s}_w^{(i)} \sim \text{Buffer}(\boldsymbol{s}_w)$
6:         Compute distances for each candidate waypoint:
    $d^{(i)} \leftarrow \log \frac{C(F=1|\boldsymbol{s}_w, \boldsymbol{s}_g)}{C(F=0|\boldsymbol{s}_w, \boldsymbol{s}_g)} + \log \frac{C(F=1|\boldsymbol{s}_0, \boldsymbol{s}_w)}{C(F=0|\boldsymbol{s}_0, \boldsymbol{s}_w)})$
7:         $\boldsymbol{s}_w \propto \text{SOFTMAX}(d^{(i)})$
8:     **else**
9:         Set $\boldsymbol{s}_w \leftarrow \boldsymbol{s}_g$
10: **return** $\boldsymbol{s}_w, n_g$

See Appendix A.3 for a full derivation. Since we will eventually normalize these importance weights (over $\boldsymbol{s}_w$), we do not need to estimate $Z(\boldsymbol{s}_0, \boldsymbol{s}_g)$. We call our complete algorithm *C-Planning*, and summarize it in Alg. 1. Note that our algorithm requires changing only a single line of pseudocode: during data collection, we command the agent to the intermediate waypoint $\boldsymbol{s}_w$ instead of the final goal $\boldsymbol{s}_g$. The sampling procedure in Alg. 2 is scalable and easy to implement.

The core idea of our method is to sample waypoints enroute to the goal. Of course, at some point we must command the agent to directly reach the goal. We introduce a hyperparameter $n_g$ to indicate how many times we resample the intermediate waypoint before commanding the agent to reach the goal. This parameter is not the planning horizon, which is always set to 2.

## 5 EXPERIMENTS

Our experiments study whether C-Planning can compete with prior goal-conditioned RL methods both on benchmark tasks and on tasks designed to pose a significant planning and exploration challenge. Our first experiments use these tasks to compare C-Planning to prior methods for goal-conditioned RL. We then study the importance of a few design decisions through ablation experiments.The benefits of C-Planning, compared with prior methods, are especially pronounced on these tasks. To provide more intuition for why C-Planning outperforms prior goal-conditioned RL methods, we plot the gradient norm for the actor and critic, finding that C-Planning enjoys larger norm gradients for the critic and smaller variance of the gradient for the actor. Ablation experiments study the number of waypoints used in training. We measure the inference time of choosing actions, finding that C-Planning is up to $4.7\times$ faster than prior methods that perform search at test time. [3]

**Environments.** We investigate environments of various difficulty (visualize in in Fig. 2). The first set of environments is taken from the Metaworld suite (Yu et al., 2020), a common benchmark for goal-conditioned RL. These tasks are challenging because of the manipulation skills required to push and reorient objects. While prior goal-conditioned RL methods do solve some of these tasks (e.g., `Push` and `Reach`), they stand at the edge of the capabilities of current methods. The second set of environments, 2D navigation mazes, are designed to stress-test the planning capabilities of different methods. Successfully solving these tasks requires reasoning over long horizons of up to 50 steps. These are challenging because, unlike many benchmarks, they require non-greedy exploration: greedy strategies get stuck in local optima. Our final set of environments combine the challenges of these environments. We extend the tasks in Metaworld to involve *sequential* manipulation of *multiple* objects, similar to the tasks in prior work (Singh et al., 2020). For example, the `Obstacle-Drawer-Close` task require the robotic arm to manipulate multiple objects in sequence, first pushing an object and then opening a drawer. These tasks are difficult because the agent is given no demonstrations, no reward shaping, and no manually specified distance metrics.

**Baselines.** We will compare C-Planning to a number of baselines. C-Learning is a recent goal-conditioned RL method that does not perform planning. C-Learning performs the same gradient

---
[3] Out code is available at `https://github.com/tianjunz/c-planning`.

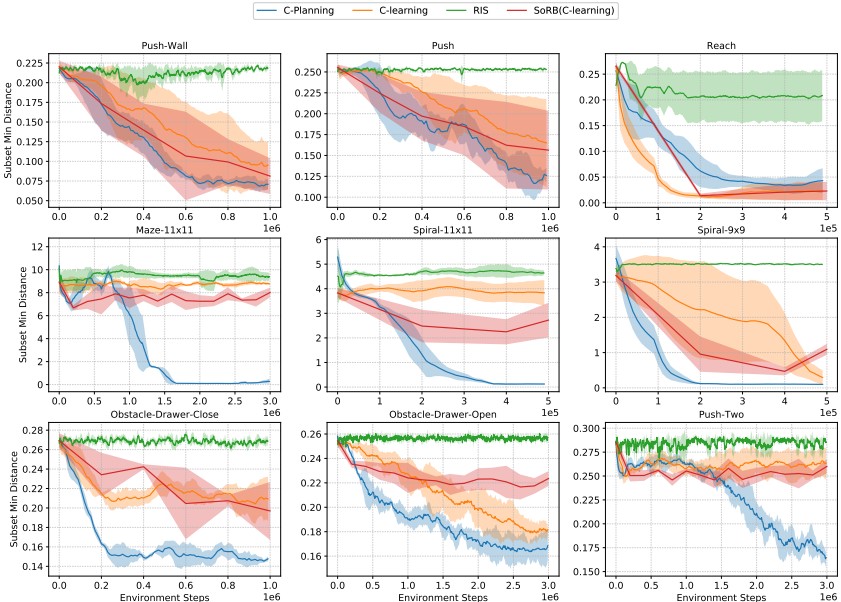

Figure 3: **Comparison of goal-conditioned RL methods**: We compare C-Planning to prior goal-conditioned RL algorithms on various tasks: *(top)* benchmark manipulation tasks from Metaworld, *(middle)* 2D maze navigation, and *(bottom)* robot manipulation tasks that require long horizon planning. For each task, we record the Euclidean distance to the goal, taking the minimum distance within an episode. All but the easiest task (Reach), C-Planning outperforms all prior methods, including those that perform planning. Only C-Planning is able to solve the most challenging navigation and manipulation tasks.

updates as C-Planning, and only differs in how experience is collected. Comparing to C-Learning will allow us to identify the marginal contribution of our curriculum of waypoints. The second baseline is SoRB (Eysenbach et al., 2019), a goal-conditioned RL method that performs search at test-time, rather than training time. While SoRB was originally implemented using Q-learning, we find that a version based on C-learning worked substantially better, so we use this stronger baseline in our experiments. Comparing against SoRB will allow us to study the tradeoffs between performing planning during training versus testing. Because SoRB performs search at testing, it is considerably more expensive to deploy in terms of computing. Thus, even matching the performance of SoRB, without incurring the computational costs of deployment would be a useful result. The third baseline is an ablation of our method designed to resemble RIS (Chane-Sane et al., 2021). Like C-Planning, RIS performs planning during training and not testing, but the planning is used differently from C-Planning. Whereas C-Planning uses planning to collect data, leaving the gradient updates unchanged, RIS modifies the RL objective to include an additional term that resembles behavior cloning. Our comparison with RIS thus allows us to study how our method for using the sampled waypoints compares to alternative methods to learn from those same sampled waypoints.

## 5.1 COMPARISON WITH PRIOR GOAL-CONDITIONED RL METHODS

To compare C-Planning to prior goal-conditioned RL algorithms, we start with three tasks from the MetaWorld suite: Reach, Push, Push-Wall. While C-Planning learns slightly slower than prior methods on the easiest task (Reach), we see a noticeable improvement over prior methods on the more challenging Push and Push-Wall tasks, tasks that require reasoning over longer horizons to solve. This observation suggests that that the waypoint sampling performed by C-Planning might be especially beneficial for solving tasks that require planning over long horizons.

To test the planning capabilities of C-Planning, we design a series of 2D navigation mazes. While the underlying locomotion motions are simple, these tasks post challenges for planning and are designed so that a greedy planning algorithm would fail. On all three mazes, C-Planning learns faster than all baselines and achieves a lower asymptotic distance to the goal, as compared to the baselines. On the most challenging maze, Maze-11x11, only our method is able to make any learning progress. Of particular note is the comparison with SoRB, which uses the same underlying goal-conditioned RL algorithm as C-Planning, but differs in that it performs search at testing, rather than training. While SoRB performs better than C-learning, which does not perform planning, SoRB consistently

Figure 4: **Planning at Training Versus Testing:** We ablate the number of intermediate waypoints used to reach the goal, and compare against a variant of our method that performs planning during both training and testing ("C-Planning + SoRB"). This variant does not improve performance, indicating that the feedforward policy learned by C-Planning has already "internalized" these planning capabilities.

performs worse than C-Planning. This observation suggests that C-Planning is not just amortizing the cost of performing search. Rather, these results suggest that incorporating planning into training can produce significantly larger gains than incorporating planning into the deployed policy.

As the last set of experiments, study higher dimensional tasks that require long-range planning. We design three environments: `Obstacle-Drawer-Close`, `Obstacle-Drawer-Open` and `Push-Two`. These tasks have a fairly large dimension (15 to 18) and require sequential manipulation of multiple objects. C-Planning outperforms all baselines on all three tasks. While C-learning makes some learning progress, SoRB (which modifies C-learning with search at test time) does not improve the results. RIS, which also samples waypoints during training but uses those waypoints differently, does not make any learning progress on these tasks. Taken together, these results suggest that waypoint sampling (as performed by C-Planning) can significantly aid in the solving of complex manipulation tasks, but only if that waypoint

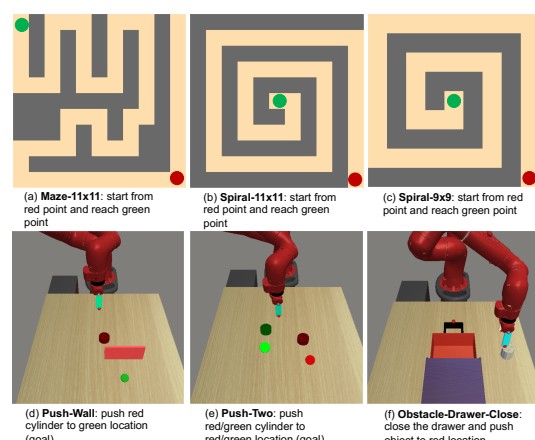

Figure 2: **Environments**: Visualization of the 2D navigation maze environments *(top row)* and robotics manipulation tasks *(bottom row)*.

sampling is performed during training. Moreover, the comparison with RIS suggests that those waypoints should be used to collect new data, rather than to augment the policy learning objective. To the best of our knowledge, C-Planning is the first method to learn manipulation behavior of this complexity without additional assumptions (such as dense rewards or demonstrations).

## 5.2 WHY DOES C-PLANNING WORK?

We ran many additional experiments to understand *why* C-Planning works so well. To start, we run ablation experiments to answer two questions: *(1)* Although C-Planning only applies planning at training time, does *additionally* performing planning at test time further improve performance? *(2)* How many times should we resample the waypoint before directing the agent to the goal?

Fig 4 shows the results of ablation experiments. C-Planning performs the same at test-time with and without SoRB-based planning, suggesting that our training-time planning procedure already makes the policy sufficiently capable of reaching distant goals, such that it does not benefit from additional test-time planning. Fig. 4 also shows that C-Planning is relatively robust to $n_g$, the number of waypoints used before directing the agent to the goal. For all choices of this hyperparameter, C-Planning still manages to solve all the tasks.

**Gradient analysis.** We provide an empirical observation why C-Planning is better than vanilla goal-conditioned RL methods. We hypothesize that, by introducing waypoints, C-Planning provides a better learning signal, increasing the gradient norm for training the critic and decreasing the variance of the gradient for the policy. Prior work has theoretically analyzed the importance of gradient norms for RL (Agarwal et al., 2021) and we will provide some empirical evidence for the importance of controlling gradient norms. In addition, variance reduction has also long been a

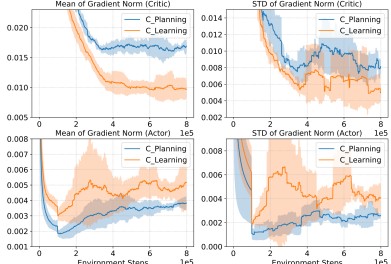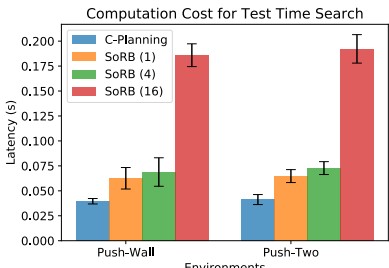

Figure 6: **Gradient Analysis and Computation Cost:** *(Left)* Mean and standard deviation of the gradient norm for the actor and critic networks. C-Planning has a larger critic gradient norm than C-Learning. C-Planning shows a smaller variance comparing to C-Learning. *(Right)* Computation cost in latency of C-Planning and SoRB with various number of waypoints.

core problem in RL (Anschel et al., 2017; Greensmith et al., 2004) and smaller variance will help stabilize the RL algorithm. In Fig. 6, we plot the norm of the gradient (mean and variance) of both C-Planning and C-Learning using the Maze-11x11 environment, finding that our method increases the norm of the critic norm and decreases the variance of the actor gradient norm.

**Test-time latency.** One important advantage of C-Planning is that it enables direct execution at test time, does not require any online planning. At test time, we directly set the agent's goal to the final goal state without commanding any intermediate states. This makes C-Planning considerably different from previous planning methods, such as SoRB (Eysenbach et al., 2019). Not only does our method achieve a higher return, but it does so with significantly less computing at deployment. We measure the execution time at evaluation time for C-Planning and SoRB with a various number of waypoints in Fig. 6. C-Planning is $1.74\times$ faster than planning over four waypoints and $4.71\times$ faster than planning over 16 waypoints in `Push-Wall` environment.

**Visualizing the training dynamics.** To further gain intuition into the mechanics of our method, we visualize

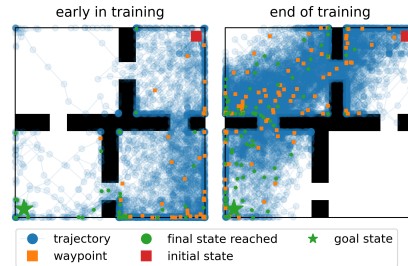

Figure 5: **Waypoint sampling:** *(Left)* Early in training, the agent samples waypoints closer to the initial state. *(Right)* At convergence, waypoints distributed along states visited by the optimal policy, as predicted by our theory.

how the distribution over waypoints changes during training of the 2D navigation of the four rooms environment. Fig. 5 shows the sampled waypoints. The value functions (i.e., future state classifiers) are randomly initialized at the start of training, so the waypoints sampled are roughly uniform over the state space. As training progresses, the distribution over waypoints converges to the states that an optimal policy would visit enroute to the goal. While we have only shown one goal here, our method trains the policy for reaching all goals. This set of experiments provides intuition of how C-Planning works as the distribution of waypoints shrinks from a uniform distribution to the single path connecting start state and goal state. This visualization is aligned with our theory (Eq. 2), which says that the distribution of waypoints should resemble the states visited by the optimal policy.

## 6 CONCLUSION

In this paper, we introduced C-Planning, a method for incorporating planning into the training of goal-conditioned agents. The method enables the automatic generation of a curriculum over intermediate waypoints. Unlike prior methods, our approach avoids the computational complexity of performing search at test time. C-Planning is simple to implement on top of existing goal-conditioned RL algorithms and achieves state-of-the-art results on a range of navigation and manipulation tasks.

Our work suggests a few directions for future work. First, C-Planning samples waypoint that correspond to the optimal state distribution, which is proovably optimal in some settings (Kakade & Langford, 2002). How might similar ideas be applied to reward-driven tasks? Second, while this paper studied sought to learn agents for reaching distant goals, we assumed that those distant goals were provided as part of the task definition. Goal-conditioned algorithms that can propose their own distant goals, while already studied in some prior work (Florensa et al., 2018; Pong et al., 2020), remains an important direction for future work.

## 7 ACKNOWLEDGEMENTS

This material is supported by the Fannie and John Hertz Foundation and the NSF GRFP (DGE1745016). In addition, this research is also supported by NSF CISE Expeditions Award CCF-1730628. UC Berkeley research is also supported by gifts from Alibaba, Amazon Web Services, Ant Financial, CapitalOne, Ericsson, Facebook, Futurewei, Google, Intel, Microsoft, Nvidia, Scotiabank, Splunk and VMware.

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

# A  PROOFS AND ADDITIONAL ANALYSIS

## A.1  DERIVING THE LOWER BOUND (PROOF OF LEMMA 1)

We first formulate the sampling procedure on starting states $s_0$, waypoints $s_w$, goals $s_g$ and the corresponding time horizon variable $t_1$ and $t_2$. Then we derive a lower bound on the target log density of Eq. 2. We then show that optimizing the lower bound through an EM procedure is equivalent to breaking the goal-reaching task into a sequence of easier sub-problems. Finally, we wrapped up this section with a practical algorithm.

**Data Generation Process.**    The generative model for which inference corresponds to our planning procedure can be formulated as follows. The episode starts by sampling an initial state $s_0 \sim p_0(s_0)$. Then it samples a geometric random variable $t_1 \sim Geom(1-\gamma)$ and roll out the policy $\pi(a \mid s, s_g)$ for exactly $t_1$ steps, starting from state $s_0$. We define $s_w$ to be the state where we end up (i.e., $s_w \triangleq s_{t_1}$). Thus, $s_w$ is sampled $s_w \sim p_{\text{GEOM}}^{\pi(\cdot \mid \cdot, s_g)}(s_{t+} \mid s_0)$. We then sample another geometric random variable $t_2 \sim Geom(1-\gamma)$ and roll out the policy $\pi(a \mid s, s_g)$ for exactly $t_1$ steps, starting from state $s_w$. We define $s_g$ to be the state where we end up (i.e., $s_g \triangleq s_{t_1+t_2}$). Thus, $s_g$ is sampled $s_g \sim p_{Geom}^{\pi(\cdot \mid \cdot, s_g)}(s_{t+} \mid s_w)$. Note that the time index of the final state $s_g$ is a sample from a negative binomial distribution: $t_1 + t_2 \overset{d}{=} NB(p = 1-\gamma, n = 2)$. We can equivalently express the sampling of $s_g$ as $s_g \sim p_{NegBinom}^{\pi(\cdot \mid \cdot, s_g)}(s_{t+} \mid s_0)$.

**Inference process.**    Under the formulation of the data generation process above, we then aim to answer the following question in the inference procedure: what intermediate states would a policy visit if it eventually reached the goal state $s_g$? Formally, we will estimate a distribution $q(s_w \mid s_0, s_g) \approx p(s_w \mid s_0, s_g)$.

We learn $q(s_w \mid s_0, s_g)$ by optimizing a evidence lower bound on our main objective (Eq. 2).

$$\log p_{\text{NEGBINOM}}^{\pi(\cdot \mid \cdot, s_g)}(s_{t+} = s_g \mid s_0) \tag{5}$$

$$= \log \int p_{\text{GEOM}}^{\pi(\cdot \mid \cdot, s_g)}(s_{t+} = s_g \mid s_w) p_{\text{GEOM}}^{\pi(\cdot \mid \cdot, s_g)}(s_w \mid s_0) ds_w \tag{6}$$

$$= \log \int p_{\text{GEOM}}^{\pi(\cdot \mid \cdot, s_g)}(s_{t+} = s_g \mid s_w) p_{\text{GEOM}}^{\pi(\cdot \mid \cdot, s_g)}(s_w \mid s_0) \frac{q(s_w \mid s_g, s_0)}{q(s_w \mid s_g, s_0)} ds_w \tag{7}$$

$$\geq \int q(s_w \mid s_g, s_0) \left( \log p_{\text{GEOM}}^{\pi(\cdot \mid \cdot, s_g)}(s_{t+} = s_g \mid s_w) + \right. \tag{8}$$

$$\left. \log p_{\text{GEOM}}^{\pi(\cdot \mid \cdot, s_g)}(s_w \mid s_0) - \log q(s_w \mid s_g, s_0) \right) ds_w \tag{9}$$

$$\triangleq \mathcal{L}(\pi, q(s_w \mid s_g, s_0)). \tag{10}$$

Note that $s_g$ is conditionally independent of $s_0$ given $s_w$, so the $p^\pi(s_{t+} = s_g \mid s_w)$ terms on the RHS need not be conditioned on $s_0$. The evidence lower bound, $\mathcal{L}$, depends on two quantities: the goal-conditioned policy and the distribution over waypoints. The objective for the goal-conditioned policy is to maximize the probabilities of reaching the waypoint and reaching the final state. The objective for the waypoint distribution is to select waypoints $s_w$ that satisfy two important properties: the current policy should have a high probability of successfully navigating from the initial state to the waypoint and from the waypoint to the final goal. Note that the optimal choice for the waypoint distribution automatically depends on the current capabilities of the goal-conditioned policy.

Before optimizing the lower bound, we introduce a subtle modification to the lower bound:

$$\mathcal{L}_2(\pi, q(s_w \mid s_g, s_0)) \triangleq \int q(s_w \mid s_g, s_0) \left( \log p_{\text{GEOM}}^{\pi(\cdot \mid \cdot, s_g)}(s_{t+} = s_g \mid s_w) + \right. \tag{11}$$

$$\left. \log p_{\text{GEOM}}^{\pi(\cdot \mid \cdot, s_w)}(s_w \mid s_0) - \log q(s_w \mid s_g, s_0) \right) ds_w. \tag{12}$$

The difference, highlighted in orange, is that the probability of reaching the waypoint is computed for a goal-conditioned policy that is commanded to reach that waypoint, rather than the final goal. We show that this new objective is also an evidence lower bound on the same goal-reaching objective

(Eq. 2), but modified such that the sequence of *commanded* goals is treated as an additional latent variable.

Assume that the initial state $s_0$ and the goal state $s_g$ are given. As before, we want to find a policy that maximizes the probability of reaching $s_g$. However, we now consider jointly optimizing over both the policy and the sequence of goals we command for that policy. We use $s_c^{(t)}$ to denote the goal commanded at time $t$. We can write this optimization problem as follows:

$$\max_{\pi, s_c^{(1:\infty)}} \mathcal{F}(\pi, s_c^{(1:\infty)}) \triangleq \log p^{\pi(\cdot|\cdot, s_c^{(1:\infty)})}(s_{t+} = s_g \mid s_0). \tag{13}$$

Applying Jensen's inequality, we obtain a lower bound that looks similar to before:

$$\mathcal{F}(\pi, s_c^{(1:\infty)}) \geq \mathbb{E}_{q(s_w|s_g, s_0)} \left[ \log p_{\text{GEOM}}^{\pi(\cdot|\cdot, s_c^{(1:\infty)})}(s_{t+} = s_g \mid s_w) + \log p_{\text{GEOM}}^{\pi(\cdot|\cdot, s_c^{(1:\infty)})}(s_w \mid s_0) - \log q(s_w \mid s_g, s_0) \right] \tag{14}$$

$$= \mathbb{E}_{q(s_w|s_g, s_0)} \left[ \log p_{\text{GEOM}}^{\pi(\cdot|\cdot, s_c^{(t_w+1:\infty)})}(s_{t+} = s_g \mid s_w) + \log p_{\text{GEOM}}^{\pi(\cdot|\cdot, s_c^{(1:t_w)})}(s_w \mid s_0) - \log q(s_w \mid s_g, s_0) \right]. \tag{15}$$

In the second line, we introduce $t_w$ as the time when the waypoint is reached. This allows us to clarify that the probability of reaching the waypoint only depends on the commanded goals through time $t_w$, $s_c^{(1:t_w)}$. This lower bound holds for any choice of the commanded waypoints, $s_c^{(1:\infty)}$. Directly optimizing for the sequence of waypoints is challenging for two reasons. First, it requires estimating the probability of commanding one goal but reaching any other state. Second, if we command one goal when trying to reach a different goal, then the goal conditioned policy may not learn to associate the commanded goal with the desired outcome. For both these reasons, we choose to not optimize the lower bound with respect to the commanded goals, but rather manually specify the commanded goals as follows:

$$s_c^{(1:t_w)} = s_w, s_c^{(t_w+1:\infty)} = s_g. \tag{16}$$

Thus, we have recovered the objective in Eq. 12. As our lower bound holds for any choice of commanded waypoint, it also holds for this choice.

## A.2 THE OPTIMAL WAYPOINT DISTRIBUTION (PROOF OF LEMMA 2

This section proves Lemma 2.

*Proof.* Recall that our goal is to solve the following maximization problem:

$$\max_{q(s_w|s_g, s_0)} \mathbb{E}_{q(s_w|s_g, s_0)} \left[ \log p_{\text{GEOM}}^{\pi(\cdot|\cdot, s_g)}(s_{t+} = s_g \mid s_w) + \log p_{\text{GEOM}}^{\pi(\cdot|\cdot, s_w)}(s_w \mid s_0) - \log q(s_w \mid s_g, s_0) \right]. \tag{17}$$

Note that the waypoint distribution must integrate to one. The Lagrangian can be written as

$$\mathbb{E}_{q(s_w|s_g, s_0)} \left[ \log p_{\text{GEOM}}^{\pi(\cdot|\cdot, s_g)}(s_{t+} = s_g \mid s_w) + \log p_{\text{GEOM}}^{\pi(\cdot|\cdot, s_w)}(s_w \mid s_0) - \log q(s_w \mid s_g, s_0) \right] + \tag{18}$$

$$\lambda \left( \int q(s_w \mid s_0, s_g) ds_w - 1 \right), \tag{19}$$

where $\lambda$ is a Lagrange multiplier. We then take the derivative with respect to $q(s_w \mid s_g, s_0)$:

$$\frac{d}{dq(s_w \mid s_0, s_g)} = \frac{-q(s_w \mid s_0, s_g)}{q(s_w \mid s_0, s_g)} + \log p_{\text{GEOM}}^{\pi(\cdot|\cdot, s_g)}(s_{t+} = s_g \mid s_w) + \tag{20}$$

$$\log p_{\text{GEOM}}^{\pi(\cdot|\cdot, s_g)}(s_w \mid s_0) - \log q(s_w \mid s_g, s_0) + \lambda \tag{21}$$

$$= -1 + \log p_{\text{GEOM}}^{\pi(\cdot|\cdot, s_g)}(s_{t+} = s_g \mid s_w) + \log p_{\text{GEOM}}^{\pi(\cdot|\cdot, s_w)}(s_w \mid s_0) - \tag{22}$$

$$\log q(s_w \mid s_g, s_0) + \lambda. \tag{23}$$

We then set this derivative equal to zero and solve for $q(s_w \mid s_g, s_0)$:

$$q(s_w \mid s_g, s_0) = e^{\lambda - 1} p_{\text{GEOM}}^{\pi(\cdot|\cdot, s_g)}(s_{t+} = s_g \mid s_w) p_{\text{GEOM}}^{\pi(\cdot|\cdot, s_w)}(s_w \mid s_0).$$

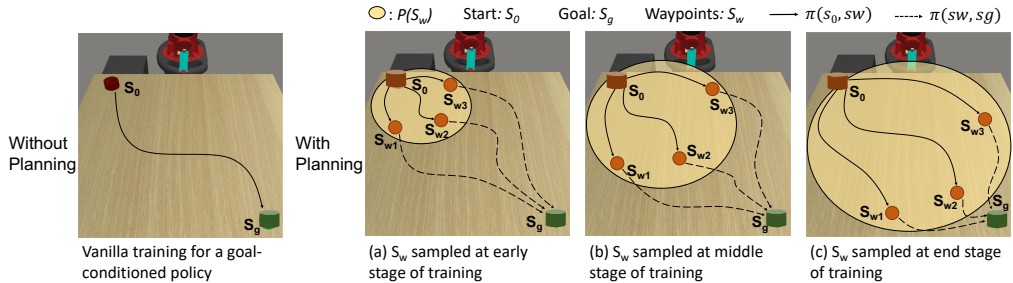

Figure 7: **C-Planning Curriculum**: Illustration of planning over waypoints distribution $p(s_w)$. Goal-conditioned RL directly commands the agent to the final goal. C-planning first samples an intermediate waypoint $s_w$ from $p(s_w)$, directs the agent to that waypoint, and then commands the final goal after the agent has reached the waypoint. Note that the $p(s_w)$ is proportional to the state density of the current policy, so the high probability region will expand from starting state $s_0$ to the goal state $s_g$, as the agent explores the environment.

Finally, we determine the value of $\lambda$ such that $q(s_w \mid s_0, s_g)$ integrates to one. We can then express the optimal waypoint distribution as follows:

$$q^*(s_w \mid s_g, s_0) = \frac{p_{\text{GEOM}}^{\pi(\cdot\mid\cdot,s_g)}(s_g \mid s_w)p_{\text{GEOM}}^{\pi(\cdot\mid\cdot,s_w)}(s_w \mid s_0)}{\int p_{\text{GEOM}}^{\pi(\cdot\mid\cdot,s_g)}(s_g \mid s'_w)p_{\text{GEOM}}^{\pi(\cdot\mid\cdot,s_w)}(s'_w \mid s_0)ds'_w}.$$

$\square$

### A.3 ESTIMATING IMPORTANCE WEIGHTS (PROOF OF LEMMA 3)

This section proves Lemma 3.

*Proof.* Define the normalizing constant as follows

$$Z(s_0, s_g) = \frac{b(s_g)}{\int p_{\text{GEOM}}^{\pi(\cdot\mid\cdot,s_g)}(s_g \mid s'_w)p_{\text{GEOM}}^{\pi(\cdot\mid\cdot,s_w)}(s'_w \mid s_0)ds'_w}.$$

Substituting $Z(s_0, s_g)$ into the RHS of Eq. 4 and simplifying the result, we show that it equals the LHS of Eq. 4:

$$\frac{C_\theta(s_w, s_g)}{1 - C_\theta(s_w, s_g)}\frac{C_\theta(s_0, s_w)}{1 - C_\theta(s_0, s_w)}Z(s_0, s_g) \tag{24}$$

$$= \frac{C_\theta(s_w, s_g)}{1 - C_\theta(s_w, s_g)}\frac{C_\theta(s_0, s_w)}{1 - C_\theta(s_0, s_w)}\frac{b(s_g)}{\int p_{\text{GEOM}}^{\pi(\cdot\mid\cdot,s_g)}(s_g \mid s'_w)p_{\text{GEOM}}^{\pi(\cdot\mid\cdot,s_w)}(s'_w \mid s_0)ds'_w} \tag{25}$$

$$= \frac{p_{\text{GEOM}}^{\pi(\cdot\mid\cdot,s_g)}(s_{t+} = s_g \mid s_w)}{b(s_g)}\frac{p_{\text{GEOM}}^{\pi(\cdot\mid\cdot,s_w)}(s_{t+} = s_w \mid s_0)}{b(s_w)}\frac{b(s_g)}{\int p_{\text{GEOM}}^{\pi(\cdot\mid\cdot,s_g)}(s_g \mid s'_w)p_{\text{GEOM}}^{\pi(\cdot\mid\cdot,s_w)}(s'_w \mid s_0)ds'_w} \tag{26}$$

$$= \frac{p_{\text{GEOM}}^{\pi(\cdot\mid\cdot,s_g)}(s_{t+} = s_g \mid s_w)p_{\text{GEOM}}^{\pi(\cdot\mid\cdot,s_w)}(s_{t+} = s_w \mid s_0)}{\int p_{\text{GEOM}}^{\pi(\cdot\mid\cdot,s_g)}(s_g \mid s'_w)p_{\text{GEOM}}^{\pi(\cdot\mid\cdot,s_w)}(s'_w \mid s_0)ds'_w}\frac{1}{b(s_w)} \tag{27}$$

$$= \frac{q(s_w \mid s_0, s_g)}{b(s_w)}. \tag{28}$$

$\square$

## B ADDITIONAL EXPERIMENTS

### B.1 AN ORACLE EXPERIMENT

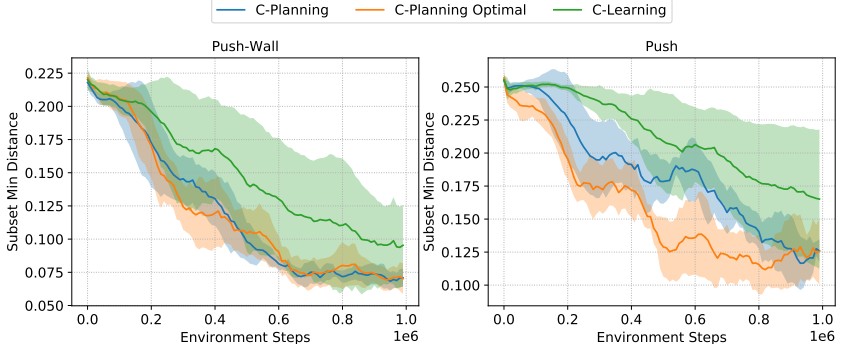

Figure 8: **Oracle experiment with the optimal generative model**: Goal-conditioned RL learns the task much faster if we can sample the waypoints from the expert policy's marginal state distribution (C-Planning Optimal). Compared to planning with optimal state density distribution, planning using a learned classifier (C-Planning) shows comparable performance.

We run an experiment to confirm the well-known result (Kakade & Langford, 2002) that the optimal initial state distribution for learning a task is the state distribution of that task's optimal policy. Intuitively, if we could sample exactly from the state distribution of the optimal policy, then an RL agent could learn to reach goals more quickly. This is not surprising as it resembles behavioral cloning of the optimal policy. In this oracle experiment, it is assumed we are given the access to an optimal policy. We achieve this by following the sampling procedure from Algorithm. 1, but replace the classifier $C$ with that learned from the optimal policy $C^*$. We then perform RL to reach

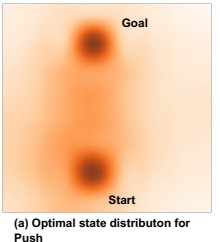

(a) Optimal state distributon for Push

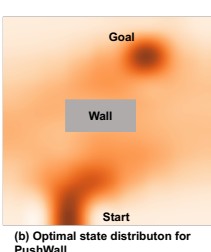

(b) Optimal state distributon for PushWall

Figure 9: *(Left)*An agent must navigate from the `start` state to the `goal` state. The heatmap visualizes the marginal state distribution of the optimal policy.

waypoints and then the final goal. Results in Fig. 8 show that using the optimal policy as the initial state distribution results in faster learning than using the original state distribution. We also visualize the state density distribution of the optimal policy to provide more qualitative intuition on how the algorithm works. Please refer to Appendix B.2 for more details.

In addition, we are also interested in measuring how well does planning through a learned model (C-Planning) performs if we don't have access to the optimal generative model. Results in Fig. 8 also show that C-Planning achieves performance comparable to planning via optimal generative model in the Push and the Push-Wall environment.

## B.2 VISUALIZATION OF STATE DENSITY MAP OF OPTIMAL POLICY

We conduct this experiment on a 2D navigation task shown in Fig. 8 (left), where we have also visualized the original initial state distribution, the state distribution of an optimal policy, and the goal state. To conduct this experiment, we apply a state-of-the-art goal-conditioned RL algorithm (C-learning) in the two settings with different initial state distributions. For fair evaluation, we evaluate the policies learned in both settings using the original initial state distribution. The results shown in Fig. 8 (right) show that starting from the optimal initial state distribution results in 2.4x faster learning.

## B.3 ABLATIONS TO HER, SKEWFIT AND SORB (C-PLANNING)

We perform additional experiments, comparing C-Planning with HER (Andrychowicz et al., 2017), SkewFit (Pong et al., 2020) and SoRB (Eysenbach et al., 2019) on C-Planning. HER barely works on only simple experiments (e.g., Reach) but fails on harder ones. Similar results are also been

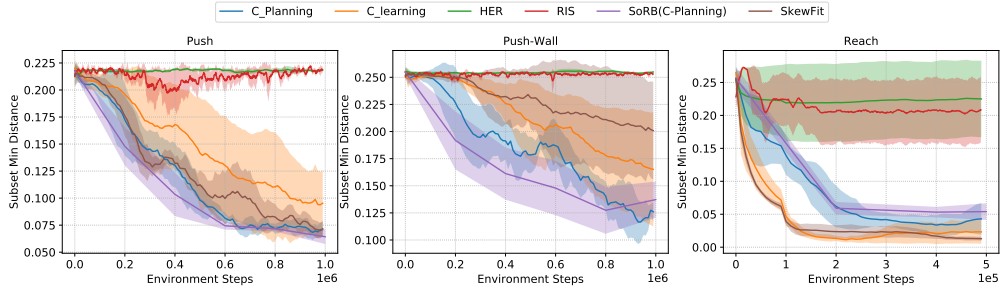

Figure 10: **Ablation with HER, SkewFit and SoRB**: Comparison of C-Planning to more baselines like HER, SkewFit and SoRB. HER only shows good performance on easy task like Reach but fails on harder ones. SkewFit shows a little better performance in Reach task but is a little worse in Push and Push-Wall. Additional ablation study on SoRB(C-Planning) shows small benefits are gained by adding SoRB at test time.

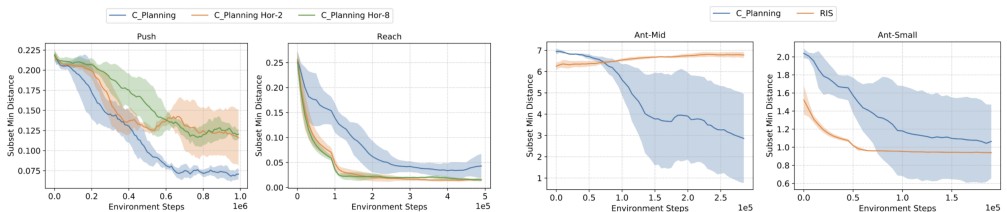

Figure 11: **Ablation on the Planning Horizon and Experiments on Ant-Maze**: (*left*) Albation study of the planning horion. A Larger planning horizon will sometimes help the performance of tasks, but the improvement is not consistent. (*right*) Experiments on small scale Ant-Maze. RIS manage to reach the goal quickly in Ant-Small environment but fails in the Ant-Mid.

observed in the C-Learning (Eysenbach et al., 2020) paper. We implement a version of SkewFit algorithm on top of our framework. The only change it did is SkewFit samples swaypoints according to the inverse of a state density model. We see that SkewFit has better performance in Reach task but fails in PushWall task. We also compare C-Planning with SoRB(C-Planning) which does SoRB at test time. Results show that SoRB(C-Planning) performs as well as C-Planning at test time.

## B.4 ABLATIONS ON PLANNING HORIZON

We perform an ablation study on the planning horizon (number of waypoints used when doing sampling). We implement this in a way of iterative planning: take horizon of 4 for example, we first planning the middle point $s_{w2}$ via starting point and goal of $(s_0, s_g)$; then planning $s_{w1}$ via starting point and goal $(s_0, s_{w2})$. Results show that planning for a more fine-grained fashion (larger number of waypoints) is not always helping. In the Fig. 11 left, in Reach task, planning over more waypoints helps but it decreases the performance in Push task. Note that here the number of waypoints is used for planning, while in Fig. 4 the number of waypoints is how many waypoints to reach before changing to the final goal. These two are different quantities.

## B.5 ADDITIONAL EXPERIMENTS ON ANT-MAZE

We performance an additional experiments in the Ant-Maze environment. We design a very simple environment: command the ant to a specific goal position with a short (Ant-Small) or a long (Ant-Mid) distance. The distance between starting point and goal is 2.5 and 7.5 repectively for Ant-Small and Ant-Mid. The only two modifications we did comparing to the original RIS (Chane-Sane et al., 2021) is change the maximum time horizon to 300 (RIS uses max time steps of 600) and reset the agent to a fixed area (RIS resets the position of the agent uniformly). In Fig. 11 right, RIS achieves better performance in Ant-Small environment while fails in Ant-Mid environment. Note that C-Planning doesn't use any termination function and reward function while RIS heavily relies on them.

We also study the performance of RIS under these design choices. We mainly concerned with two settings: random initialize the agent's position versus fixed initialize the agent's position and choosing a termination threshold of 1.0 versus 0.5. In Fig. 12, we can clearly see the results that using a slightly different threshold factor for termination will greatly affect the performance of RIS: using 1.0, the agent almost fails to learn any policy; while using 0.5, the agent is learning very fast. This shows that RIS is very sensitive to the choice of this termination function. It again proves the benefit of C-Planning since we don't rely on any reward function and termination function.

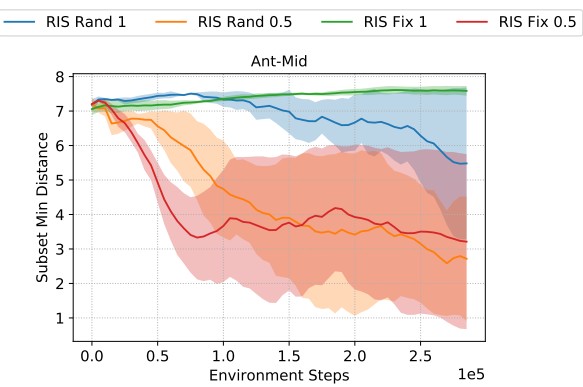

Figure 12: **Ablations on design components of RIS:** We study design factors that might affect the performance of RIS. Different threshold on termination function (1.0 versus 0.5) is affecting the performance of RIS significantly.

## C   EXPERIMENTAL DETAILS

In this section, We provide the essential hyperparameters for reproducing our experiments in this section. We also introduce the hyperparameters used in baselines and provide a detailed description of environmental design.

### C.1   IMPLEMENTATION DETAILS

We introduce the hyperparameters used in C-Planning. Note that in C-Learning, only a classifier on $(\boldsymbol{s}_t, \boldsymbol{s}_{t+}, \boldsymbol{a}_t)$ is needed. In C-Planning, in order to sample waypoints from the distribution, an additional classifier on $(\boldsymbol{s}_t, \boldsymbol{s}_{t+})$ is needed. We also introduce two hyperparameters: Maximum Steps Reaching Goal ($N_g$ in Alg.1) forces the agent to change the intermediate goal if the original goal hasn't been reached for some steps; Distance Threshold Reaching Goal ($\epsilon_d$ in Alg.1) for determing whether a goal is reached or not. The rest are the standard hyperparameters for SAC algorithm and we list here for reference.

Table 1: Hyperparameters used for C-Planning in all the environments in MetaWorld.

|  | Hyperparameter Value |
| --- | --- |
| Actor lr | 0.0003 |
| Action-State Critic lr | 0.0003 |
| State Critic lr | 0.00003 |
| Actor Network Size | (256, 256, 256) |
| Critic Network Size | (256, 256, 256) |
| Maximum Steps Reaching Goal | 20 |
| Distance Threshold Reaching Goal | 0.05 |
| Actor Loss Weight | 1.0 |
| Critic Loss Weight | 0.5 |
| Discount | 0.99 |
| Target Update Tau | 0.005 |
| Target Update Period | 1 |
| Number Waypoints | 5 |
| Goal Relabel Next | 0.3 |
| Goal Relabel Future | 0.2 |

Note that we use a slightly different hyperparameters for 2D maze environment and we list below, all the other hyperparameters remains the same. Note that we follow the goal relabeling changes by C-Learning (Eysenbach et al., 2020).

Table 2: Hyperparameters used for C-Planning in all the environments in 2D navigation maze.

|  | Hyperparameter Value |
| --- | --- |
| Distance Threshold Reaching Goal | 1.0 |
| Number Waypoints | 8 |
| Goal Relabel Next | 0.5 |
| Goal Relabel Future | 0.0 |

## C.2 ENVIRONMENTS

We follow the envioronment design of (Eysenbach et al., 2020) with only one noticeable difference: in the original environments of C-Learning, for the ease of training, the author set the initial position of objects to be relatively near the arm so the arm can easily push the object, getting a better learning signal. We intentionally set the initial state of object to be far away from the arm. This significantly increase the difficulty of learning. We'll release these environment with the code.

Table 3: Max number of time steps used for each environment.

|  | Max Time Steps |
| --- | --- |
| Spiral 9x9 | 200 |
| Spiral 11x11 | 200 |
| Maze 11x11 | 200 |
| Sawyer Push | 50 |
| Sawyer Reach | 50 |
| Sawyer Push-Wall | 50 |
| Obstacle-Drawer-Close | 150 |
| Obstacle-Drawer-Open | 150 |
| Sawyer Push-Two | 150 |

## C.3 BASELINES

We also provide the hyperparameters associated with the baselines. The two baselines we care about: C-Learning and RIS. We summarize their hyperparameters in the table:

Table 4: Hyperparameters used for C-Learning in all the environments in MetaWorld.

|  | Hyperparameter Value |
| --- | --- |
| Actor lr | 0.0003 |
| Action-State Critic lr | 0.0003 |
| Actor Network Size | (256, 256, 256) |
| Critic Network Size | (256, 256, 256) |
| Actor Loss Weight | 1.0 |
| Critic Loss Weight | 0.5 |
| Discount | 0.99 |
| Target Update Tau | 0.005 |
| Target Update Period | 1 |
| Goal Relabel Next | 0.3 |
| Goal Relabel Future | 0.2 |

## D VISUALIZATION OF THE LEARNED POLICY

In order to more intuitively visualize the behavior of the learned policy and emphasize the importance of the difficulty of the learning task, we plot the visualization of our learned policy for several snap shots in an episode for environment of `Push-Two` and `Obstacle-Drawer-Open`. We see that our method successfully guide the agent to learn the behavior of push the green object first, the push the object; And first open the drawer then push the object to the desired location. We would

Table 5: Hyperparameters used for RIS in all the environments in MetaWorld.

|                     | Hyperparameter Value |
| ------------------- | -------------------- |
| epsilon             | 0.0001               |
| Replay Buffer Goals | 0.5                  |
| Distance Threshold  | 0.05                 |
| Alpha               | 0.1                  |
| Lambda              | 0.1                  |
| H lr                | 0.0001               |
| Q lr                | 0.001                |
| Pi lr               | 0.0001               |
| Encoder lr          | 0.0001               |

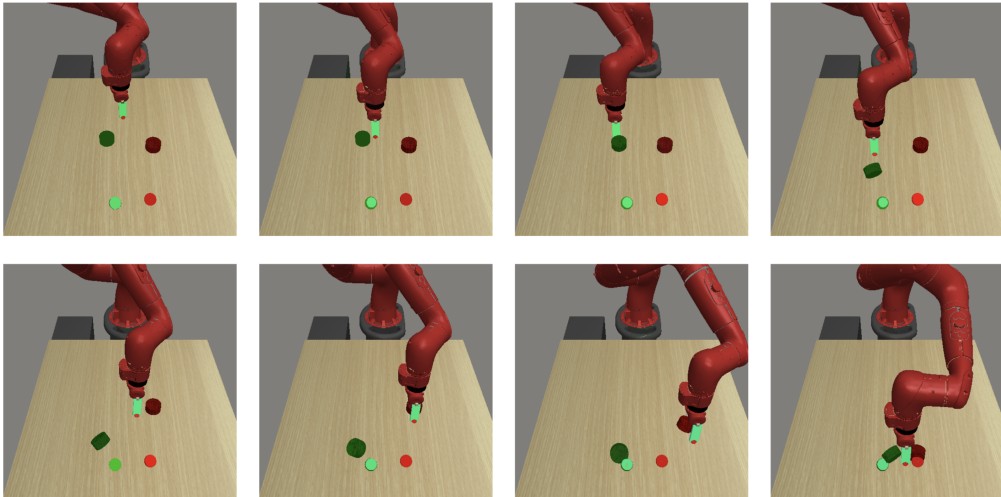

Figure 13: Visualization of the trained policy for the `Push-Two` environment. Our method successfully trains the agent to manipulate the two object in the sequential manner without any offline data, expert demonstration and reward shaping. Prior baselines all fail to demonstrate such behavior.

like to emphasize that prior baselines all fail to demonstrate such behavior without any offline data, expert demonstration and reward shaping. The successfully learning of such behavior enables the robot to do more complex tasks without any human intervention.

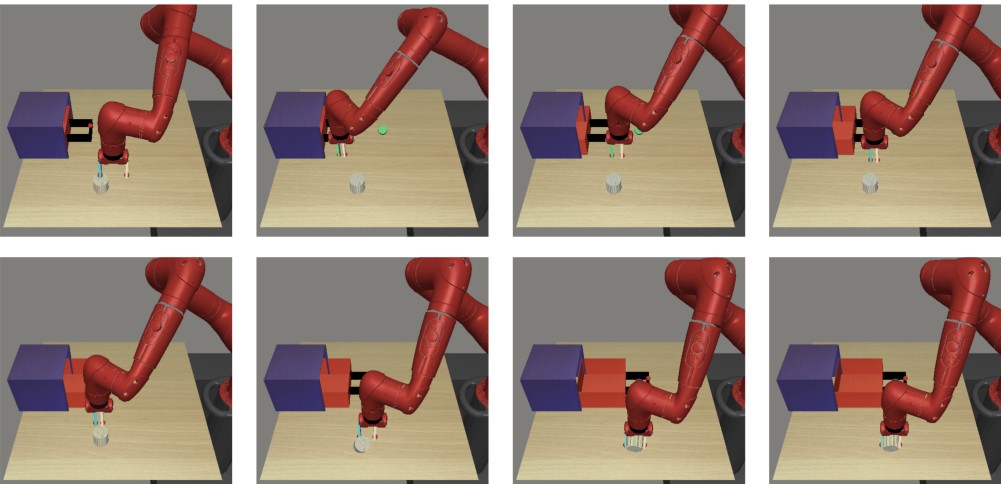

Figure 14: Visualization of the trained policy for the `Obstacle-Drawer-Open` environment. Our method successfully trains the agent to first close the drawer and then push the object to the desired location. We emphasize that such behavior is hard to obtain and most of the prior methods only manage to close the drawer.

