# OpenReview forum: "C-Planning: An Automatic Curriculum for Learning Goal-Reaching Tasks"
_ICLR.cc/2022/Conference — ICLR 2022 Poster_

### Official Review · Reviewer_1xhY · 2021-11-02

**Correctness:** 3
**Technical Novelty And Significance:** 3
**Empirical Novelty And Significance:** 2
**Recommendation:** 6
**Confidence:** 4

**Main Review:**

Main concerns
1. Missing important baselines\
The experiment is missing an important baseline: [Huang et al., 2019]. In section 2, authors claim that [Huang et al., 2019] does not perform planning in train time and does not improve the underlying goal-conditioned policy. However, this is not true. [Huang et al., 2019] also improves the quality of the data used to train the policy by improving the exploration in training. Specifically, [Huang et al., 2019] periodically populates the landmarks using the farthest-point-sampling to explore the frontier of the explored state space.
Another important baseline to consider is the goal-relabeling methods because they also provide additional learning signals from the states visited during training for GCRL. It would be very helpful to demonstrate that the proposed method is a better way to provide additional learning signals to GCRL than the alternatives.

2. More intuition on “iterative” waypoint sampling\
To my understanding, the optimal way point $s_w$ that minimizes the proposed objective is the midpoint between current $s$ and the goal state $s_g$ regardless of the choice of $n_g$. However, this is not really the optimal choice for the GCRL. The better waypoints would be (arguably) the equally-spaced $n_g$ points on the route from initial state to the goal state. If authors can share the philosophy behind this specific design choice, it would be very helpful for the reader.

3. The choice of prior\
Authors proposed to choose a specific prior model: negative binomial distribution with $p=1-\gamma$ and $n=2$. It seems this assumption is necessary for the decomposition of Eq.(2) into Eq.(4), but the specific design choice could be justified/explained better. Also, it is counter-intuitive in that the pdf of negative binomial distribution is bell-shaped. It means the agent receives a *smaller reward* if it achieves the goal “too early”.

Minor concerns

4. Searching at training time\
It may be controversial whether C-planning performs (implicit) searching in train time. The C-planning queries the distance between several pairs of states, but it does not perform any path finding

5. RIS is not learning at all\
In figure 4, the performance of RIS is consistently the worst among the compared methods and seems RIS is not learning at all even in the easiest task. Is this expected? It is unclear whether the RIS is correctly reproduced and the hyperparameters are appropriately tuned.

6. Performing the test-time search\
It seems natural to me to consider also performing the waypoint sampling in test-time similar to training. In figure 5, authors instead implemented the C-Planning + SoRB. Is there a reason that C-planning cannot perform test-time waypoint sampling? If possible, does performing test-time search improve the performance?

Questions/suggestions

- Q: in line 7 of Algorithm 1: What’s the difference between $\tau_1$ and $\tau_2$? Why do you need those two?
- Gradient analysis: it would be great if authors can provide at least an intuitive explanation of why increasing  the norm of critic norm and decreasing the variance of the actor gradient norm is helpful for learning in the paper for the readers

[1] Mapping state space using landmarks for universal goal reaching. Huang et al., 2019.

[2] Hindsight experience replay. Andrychowicz et al., 2017


**Summary Of The Paper:**

This work extends the previous work, C-learning, by using search at training time to generate a curriculum of intermediate states enroute to the goal state. The main idea is to decompose the goal reaching problem into a sequence of easier tasks of reaching the reachable intermediate states. Authors prove that under the negative binomial prior with $n=2$, the goal-reaching RL objective is lower-bounded by the sum of two goal-reaching objectives: reaching waypoint $s_w$ from the initial state $s_0$ and reaching the goal state $s_g$ from the way point $s_w$. In practice, authors proposed to uniformly sample the way points from the replay buffer and apply importance sampling afterwards to avoid the challenge of sampling high-dimensional waypoint states. The experiment was conducted on various goal-conditioned RL tasks on 2D mazes and the robotics manipulation tasks. The C-planning method outperforms the compared baselines on challenging tasks. Authors also conducted various ablation studies justifying several design choices made in the proposed method and demonstrating the light deployment-time computation cost of the proposed method.

**Summary Of The Review:**

The paper presents a non-trivial extension of the previous work and presents an interesting viewpoint of formulating implicit train-time searching as variational inference. The paper is clearly written and easy to follow. However, I found that the experiment section is relatively weak and the presented results are less convincing.

---

> ### Author Response · Authors · 2021-11-18
> **Author Response**
>
> We thank the reviewer for their review and thoughtful feedback. The reviewer seems mainly concerned about missing baselines on goal-relabeling algorithms as well as [Huang et al., 2019]. The reviewer also asks for more intuition on negative binomial distribution and iterative sampling. We have provided baselines of HER, results on SoRB combined with C-Planning. We also clarified some intuition on the negative binomial sampling.  We respond to the points brought by the reviewer below:
>
> **Q: Missing baselines on [Huang et al., 2019] and other goal-relabeling baselines.**
>
> **A:** We thank the reviewer for pointing out these baselines. We have added an ablation study comparing C-Planning with HER [1] and updated the paper accordingly (green line in Fig. 11, Appendix. B.3). We observe that HER solves a very easy task Reach in 1 out of 4 random seeds but fails on others. Similar observations can also be found in C-Learning [2].
>
> Please note that the algorithm we are comparing to C-Learning [2] is also a goal-relabeling algorithm that has a better performance compared to HER.  Regarding [Huang et al., 2019], it is somewhat hard to use their code in our setting. [Huang et al., 2019] requires a reward function and a termination function, which is different from our setting. We have already added a paragraph of discussion in Paragraph 3 Section 5. We think that it shares a lot of similarities with SoRB, which we already compared in our paper. We’ll keep trying to adapt it to our settings and run an experiment comparing C-Planning with [Huang et al., 2019] after the author response period.
>
> **Q: The proposed method is not really the optimal choice for the GCRL. The better waypoints would be (arguably) the equally-spaced ng points on the route from initial state to the goal state. If authors can share the philosophy behind this specific design choice, it would be very helpful for the reader.**
>
> **A:** We agree that our use of "optimal" in the paper (especially in Lemma 2) was unclear, and we have revised the paper to clarify what we mean by "optimal". The waypoint distribution derived in Lemma 2 is optimal in the sense that it maximizes the lower bound in Lemma 1.
>
> One useful property of our waypoint sampling is that it takes into account the capabilities of the current policy. For example, if the current policy can only successfully reach very close to the initial state, then the sampled waypoints will be near the initial state. A second useful property of our waypoint sampling is the connection with the optimal initial state distribution, discussed in the second paragraph of "Two mental models." Simple alternative strategies, such as sampling equally spaced waypoints, may not satisfy these properties.
>
> **Q: The choice of negative binomial distribution could be justified and it is counter-intuitive why the agent receives a smaller reward when reaching the goal too early.**
>
> **A:** We agree that this decision seems counterintuitive, and we have added two sentences to paragraph 2 in section 4.1 to explain this. The reasoning behind the negative binomial distribution is that we want to decompose a hard problem into a sequence of easier RL problems. Each of these easier (goal-conditioned) RL problems uses geometric distributions, and the sum of geometric random variables is a negative binomial distribution. Note that the negative binomial distribution is only used to derive our algorithm, not in the evaluation.
>
> **Q: It is controversial whether C-Planning performs search at training time.**
>
> **A:** We have removed the claim that C-Planning implicitly learns to perform search at training time.
>
> **Q: RIS is not learning at all. Is this expected?**
>
> This is not surprising as RIS was designed to solve an easier problem setting than C-planning, one where (goal-conditioned) reward functions are available to the agent. In contrast, C-planning considers the problem setting where no reward functions are given; the agent doesn't even receive any indication of when it has succeeded at reaching a particular goal. In addition to that, in our experiments, we reset the agent to be a relatively fixed position instead of randomly initialize the agent in RIS setting. Thus, this may also affect the performance of RIS.
>
> To further compare C-Planning with RIS, we ran two additional experiments, comparing both method on the easier problem setting considered in RIS in Appendix A.5, where reward functions are provided. In this setting, we can directly compare the original implementation of RIS (which makes use of reward functions) to C-Planning (which doesn't use rewards). Both RIS and C-Planning learn the easier task, but RIS learns faster. This is not surprising, as RIS is given more information than C-Planning (i.e., it can observe the reward function). Only C-Planning solves the more challenging task, and it does so even without access to a reward function.

---

> > ### Author Response · Authors · 2021-11-18
> > **Author Response**
> >
> > **Q: How about C-Planning performing SoRB at test time?**
> >
> > **A:** We thank the reviewer for pointing this out. We have added experiments to the paper regarding this (purple line in Fig. 11, Appendix. B.3). We found combining SoRB with C-Planning does not further improve C-Planning.
> >
> > **Q: Differences between $\tau_1$ and $\tau_2$ at Alg.1 Line 7?**
> >
> > **A:** We thank the reviewer for bringing this up. It was a typo. $\tau_1$ and $\tau_2$ are indeed different trajectories: the former is the trajectory from the starting point $s_0$ to the waypoint $s_w$ and the latter is from the waypoint $s_w$ to the goal $s_g$. We have updated the paper accordingly.
> >
> > **Q: Why is the increasing norm of critic loss and decreasing variance of actor loss helpful?**
> >
> > As we stated in Section 5.2, we just want to provide a hypothesis that why C-Planning works well. This hypothesis is that: by having a lower variance of the actor loss, we can have a more stabilized training algorithm; by having a higher norm of a critic loss, we have more signals to train the goal-reaching algorithm. Much prior work has studied variance reduction of the actor (i.e., the policy gradient), finding that reducing this gradient variance leads to higher returns [3, 4]. More recent theoretical work argues higher Q value gradients will accelerate the training [5]. We have added this discussion to Paragraph 2 in Section 5.2.
> >
> > **References:**
> >
> > [1] Andrychowicz, Marcin, et al. "Hindsight experience replay." arXiv preprint arXiv:1707.01495 (2017).
> >
> > [2] Eysenbach, Benjamin, Ruslan Salakhutdinov, and Sergey Levine. "C-learning: Learning to achieve goals via recursive classification." arXiv preprint arXiv:2011.08909 (2020).
> >
> > [3] Greensmith, Evan, Peter L. Bartlett, and Jonathan Baxter. "Variance Reduction Techniques for Gradient Estimates in Reinforcement Learning." Journal of Machine Learning Research 5.9 (2004).
> >
> > [4] Anschel, Oron, Nir Baram, and Nahum Shimkin. "Averaged-dqn: Variance reduction and stabilization for deep reinforcement learning." International conference on machine learning. PMLR, 2017.
> >
> > [5] Agarwal, Alekh, et al. "On the theory of policy gradient methods: Optimality, approximation, and distribution shift." Journal of Machine Learning Research 22.98 (2021): 1-76.

---

> > > ### Comment · Reviewer_1xhY · 2021-11-26
> > > **Thank you for the rebuttal**
> > >
> > > Thank you for the rebuttal
> > >
> > > The new ablation study results (B.3) demonstrate that the C-Planning method works well compared to SkewFit and HER. And the comparison with SoRB at test time shows that adding test-time planning can provides only a small benefits. Also, authors clarified the benefit of the proposed waypoint over the equally spaced waypoints. Together, these results alleviated my concerns on missing comparison with [Huang et al., 2019] and clarified the benefit / intuition behind the proposed methods. Overall, the rebuttal made the paper more convincing.

---

> ### Author Response · Authors · 2021-11-22
> **Author Respose**
>
> We thank the reviewer for their time, their review and thoughtful feedback.
>
> We believe that our rebuttal and the revisions to the paper address all the concerns raised by the reviewer. It would be great if the reviewer could confirm that the concerns have been addressed, or clarify the concerns so that we can work to address the concerns. It would be greatly appreciated!

---

### Official Review · Reviewer_qxwP · 2021-11-02

**Correctness:** 3
**Technical Novelty And Significance:** 3
**Empirical Novelty And Significance:** 3
**Recommendation:** 8
**Confidence:** 4

**Main Review:**

## Strengths
- The presented method steps on C-Learning, but I also find it manages to introduce sufficient theoretical and practical novelty. This is primarily expressed in the waypoint idea (coined C-planning), which is very nicely motivated through the probabilistic framework. The proposed ELBO reformulation of the objective in Lemma 1 appears natural and sound to me.
- Using the classifier of C-learning to define the approximate posterior over waypoints through importance sampling is a nice conceptual trick. The outputs of the classifier are a proxy for the future state distribution, needed to evaluate the "likelihood" (waypoint-to-goal) and "prior" (start-to-waypoint) for the reweighting, so I find the usage very fitting, assuming the classifier really captures the distributions in question.
- The considered environments are sufficiently complex, albeit simulated. The proposed algorithm seems to perform well in all of them and outperforms the baselines, including C-learning and a baseline that incorporates planning during test time, which is nice to see.
- The E step of the developed EM algorithm (MC-estimation of the ELBO w.r.t. samples from the waypoint distribution) seems rather generic, and can probably be combined with approximate control algorithms other than C-learning (the M step), as long as they subscribe to the VI / probabilistic view on control.

## Weaknesses
- The ELBO (and respective PGM) in Lemma 1 is derived under the goal-conditioned policy, but later in Lemma 2, when the approximate posterior over waypoints is defined, and for pretty much all later derivations in the paper we see a goal-conditioned policy used for the waypoint-to-goal term ("likelihood" in the PGM for the ELBO), and a waypoint-conditioned policy for the start-to-waypoint term ("prior" in the PGM for the ELBO). This seems discrepant with the ELBO derivation. I checked Appendix A.1, and it hints at that, promising a proof of why this would be OK, but I did not manage to find it. I am setting my correctness score to 3 because of this issue, until it is clarified.
- I think further clarity about the exact conditional inputs fed into the C-learning classifier are necessary, preferably in the main body of the paper. E.g. appendix E.1 mentions a secondary classifier is needed, leaving the control input out (which I imagine is necessary for e.g. the weights in Lemma 3, etc.). Given that the classifier (critic) introduced in C-Learning is a proxy for the assumed distribution over future reachable states, I think this is important. In general, spelling out exactly which distributions are approximated through the classifier (and for which policies) would make the paper more self-contained.
- It was hard for me to glean from the paper why exactly RIS [2] (which adds planning to the objective directly) fails in comparison to C-planning (in which planning is expressed through the waypoint selection when collecting experience, i.e. it's reflected in the empirical of collected data). Section 2 mentions this is because "(C-planning) avoids favoring the learned policy", can you please elaborate? I think this is an important distinction.

## Further remarks & questions
- In Algorithm 1, line 7 currently says that two rollouts are added to the data set, both start-to-waypoint under the current policy. Shouldn't one of them be waypoint-to-goal, to match the second term in the ELBO, or am I misinterpreting? Also, it might be good to specify the conditioning of the policy at this point, for clarity.
- Looking at Algorithm 1, the same policy and classifier appear in every loop iteration, for different episodes. Are their parameters kept in-between episodes, or do they get reset? Also, does C-learning run to convergence for every episode? Would be good if this is specified somewhere.
- The assumed NegBinom model on the assumed distribution of episode lengths seems like a choice out of convenience. Judging by the appendix, it's equivalent to concatenating two rollouts with geometric-distributed length, which then fit the C-Learning assumption. Was there any other motivation behind it, and did you consider alternatives?
- The link to code advertised in the paper did not work for me (I only see the paper title on the page, no videos either).
- The title of section 4 is currently at the bottom of page 3.
- I think the caption of Figure 4 does not reflect the order of the figures.
- In section 5, Results, it says C-planning performs worse on Push, but if I am reading figure 4 correctly it should be Reach.

## References
[1] Benjamin Eysenbach, Ruslan Salakhutdinov, and Sergey Levine. C-learning: Learning to achieve goals via recursive classification. ICLR, 2020.

[2] Elliot Chane-Sane, Cordelia Schmid, and Ivan Laptev. Goal-conditioned reinforcement learning with imagined subgoals. ICML, pp. 1430–1440. PMLR, 2021.


**Summary Of The Paper:**

The paper presents a method for goal-conditioned RL, tailored to solving tasks with distant goals (compared to the current SotA in goal-conditioned RL). For that, a prior approach called C-Learning [1] (a VI-flavored reformulation of approximate control for goal-reaching tasks) is extended with a scheme for experience collection (collecting on-policy rollouts), which is akin to search / planning in the sense that the overall start-to-goal reaching task is broken up into collecting start-to-waypoint and waypoint-to-goal trajectories. It is argued that breaking up the problem, and respectively introducing a posterior distribution over waypoints, enables solving long-distance reaching tasks & makes for more efficient and stable goal-conditioned RL (intermediate tasks are easier to learn from, better quality of the data, waypoints easier to reach, etc.). An additional benefit is that there is no need for search during test time (faster evaluation of the policy). The claims are empirically validated on a number of relatively complex, simulated tasks (2D maze navigation, Metaworld robot arm manipulation). The method compares favorably to the considered baselines.

**Technical details**

In technical terms, the scheme starts with the assumptions in C-learning, casting the goal-reaching objective as maximizing the prob. of the goal state under the distribution over reachable future states under policy (aligned with the VI view on optimal control). From there, a latent variable for intermediate waypoints is introduced, which leads to the formulation of an ELBO for the goal-reaching objective of C-Learning. Optimizing the ELBO is done via EM: in the E step an approximate posterior distribution over the waypoint $q(s_w)$ is implicitly defined via importance-sampling (proposal is the previously-collected empirical over states), cleverly utilizing the outputs of the classifier of C-learning for the importance weights. The waypoints sampled from $q(s_w)$ then guide the sampling of on-policy rollouts (start-to-waypoint, waypoint-to-goal) that reflect the two relevant terms in the ELBO (waypoint-to-goal corresponds to the "likelihood / reconstruction" term, start-to-waypoint to the "prior" part of the KL term). Conveniently, the two terms in the ELBO can be equated to two C-learning objectives, so the C-learning algorithm can be used as a black-box on the collected rollouts (this represents the M step of EM, maximizing w.r.t. the policy & classifier parameters in expectation over the rollouts from the E step).


**Summary Of The Review:**

I find the proposed method to be a nice step-up from existing prior work, both theoretically and in terms of performance. The proposed incorporation of planning (in terms of waypoints) appears well-justified, and seems to enable solutions to a variety of goal-reaching tasks that are not trivial. Heuristics also seem to be mostly avoided, which is appreciated.

I would recommend acceptance.

---

> ### Author Response · Authors · 2021-11-18
> **Author Response**
>
> We thank the reviewer for their positive review and thoughtful feedback. The reviewer’s main concern is about the derivation of ELBO, experimental details, and why RIS works poorly. We update the ELBO in Appendix A.1, clarified the experimental details, and provided several reasons why RIS doesn’t work.  We respond to the points brought by the reviewer below:
>
> **Q: The derivation of Lemma. 1 is goal-conditioned policy, but in later parts it is waypoints-to-goal.**
>
> **A:** We thank the reviewer for pointing this out. We have updated the paper in Appendix A.1. The core idea is to treat the command goal $s_c$ as an additional variable and derive another evidence lower bound.
>
> **Q: Please specify what are the inputs to the two classifiers needed.**
>
> **A:** We adopt an approach similar to C-Learning, which is an actor-critic algorithm. So one critic function (classifier) $C_\theta(s, a, s^\prime)$  is used to train the RL agent. The other one is what we derived in the paper $C_\phi(s, s^\prime)$. We have added these details to Appendix C.1.
>
> **Q: Why does RIS fail in comparison to C-Planning?**
>
> **A:** We think that one reason could be that RIS doesn’t use waypoints sampling for exploration. Thus, the training data C-Planning collects could have higher quality. Another noticeable reason is that C-Planning works in a regime without any reward function as well as the termination function. This setting allows the full automation of the agent without any human intervention. In RIS, the environments are all designed with a reward function or a termination function, which is some limitation of it.
>
> **Q: Clarification on Alg.1 Line 7.**
>
> **A:** We thank the reviewer for bringing this up. It was a typo. The line 7 in Alg. 1 is in fact two part of a trajectory, $\tau_1 \sim p^\pi(\tau | s_0, s_w)$ and $\tau_2 \sim p^\pi(\tau | s_w, s_g)$. We have fixed this small typo and change them to $\tau_1 \sim p^\pi(\tau | s_0, s_w)$ and $\tau_2 \sim p^\pi(\tau | s_w, s_g)$.
>
> **Q: Are their parameters kept in-between episodes, or do they get reset?**
>
> **A:** The parameters of the policy and the classifier are updated at the end of each episode (L9 in Alg 1).
>
> **Q: Also, does C-learning run to convergence for every episode?**
>
> **A:** As with other RL algorithms, the episode terminates after a fixed number of time steps. For example, the Reach environment always terminates after exactly 50 steps (see Appendix C.2 for details).
>
> **Q: Do we have any motivation for using negative binomial distribution?**
>
> **A:** We agree that this decision seems counterintuitive, and we have added two sentences to paragraph 2 in section 4.1 to explain this. The reasoning behind the negative binomial distribution is that we want to decompose a hard problem into a sequence of easier RL problems. Each of these easier (goal-conditioned) RL problems uses geometric distributions, and the sum of geometric random variables is a negative binomial distribution. Note that the negative binomial distribution is only used to derive our algorithm, not in the evaluation.
>
> **Q: Link for the code doesn’t work.**
>
> **A:** We apologize for that. We are working on publishing the code at the moment and will make sure to release it in the next few days.
>
> **Q: Several typos: Title of Section 4 at the bottom of page 3. Caption if Fig. 4 doesn’t reflect the order. In Section 5, C-Planning performs worse on Push. Should be Reach.**
>
> **A:** We thank the reviewer for pointing this out. We have fixed these typos.

---

> > ### Author Response · Authors · 2021-11-24
> > **Have the revisions addressed all the reviewer's concerns?**
> >
> > Dear Reviewer,
> >
> > Thank you for raising a number of excellent points in the review. Revising the paper to address these issues has made the paper more clear and precise. We would really appreciate a reply as to whether our response and clarifications have addressed the issues raised in the review, or whether there is anything else we can address.

---

> > > ### Comment · Reviewer_qxwP · 2021-11-26
> > > **Response to rebuttal**
> > >
> > > Thank you for the rebuttal.
> > >
> > > The added derivation to appendix A.1 looks O.K. and should help with reproducibility & understanding, it is appreciated. The rest was already discussed in the response to my review. My score was already positive and I intend to maintain it.
> > >
> > > A few more points after checking the last revision (these are more minor, but if it comes to a camera-ready you might want to consider them):
> > > - Please check Algorithm 1 again, it still says $\tau_2 \sim p^\pi(\tau \mid \mathbf{s}_0, \mathbf{s}_w)$.
> > > - I'd really appreciate if you explicitly point out the different policy conditioning in the line after Lemma 2, when you reference the derivations in A.1. I believe this would help anyone who's trying to follow the exact assumptions.
> > > - In section 3, "C-Learning" there's an abrupt change in the conditioning of the classifier when you introduce equation 3 (i.e. $\mathbf{a}_t$ is not in the conditional any more). This was already addressed in our discussion, maybe consider pointing it out to the reader in the main text.

---

> > > > ### Author Response · Authors · 2021-11-29
> > > > **Thanks for the Response**
> > > >
> > > > We thank the reviewer for the response. We'll check the paper again and make changes accordingly in our next revision.

---

### Official Review · Reviewer_rAmA · 2021-11-02

**Correctness:** 4
**Technical Novelty And Significance:** 3
**Empirical Novelty And Significance:** 3
**Recommendation:** 8
**Confidence:** 5

**Main Review:**

Strength:

 Robust, solid and strong theoretical development of an EM algorithm to learn long-horizon goal conditional policies that learn in a curriculum manner.
 Clear experiments.
 Clear language and well explained paper.

Some questions :

Why the negative binomial distribution assume a n = 2 (sentence before  to equation 1) Why the number of failures is 2? Or in other terms, why the planning horizon is always 2? Because of intermediate goal and goal?  Its assuming that need to do 2 steps at least?
The criticism about other papers using a Geometric distribution seems not adequate. The interpretation of Geometric distribution in this other papers seems similar to a NegBinomail(1- lambda, n). Sg should be passed to the Algorithm 2 - C-Planning.

Weakness:

Experiments:
    1. Why experiments do not contained the same as RIS? Just for completeness to show that you do better than RIS on their experiments.
     2. Why you don't consider to compare to the Skew-fit algorithm? Even if its different there some similarity on the curriculum learning with distribution that keep being modified?

**Summary Of The Paper:**

The paper presents an goal conditioned algorithm that uses an Expectation Maximization  framework. The E step is a "graph search", and the M step is a previous developed goal reaching RL algorithm. There is an automatic curriculum learning mechanism that comes from using variational inference. The goal-conditioned policy optimization and the method for sampling waypoints are jointly optimized using the same objective.

**Summary Of The Review:**



This is valuable progress on the field of learning goal conditioned policies. curriculum learning have been a challenge and this work shows a clear advance and solid theoretical connection.

---

> ### Author Response · Authors · 2021-11-18
> **Author Response**
>
> We thank the reviewer for their positive review and thoughtful feedback. The reviewer seems to be mainly concerned about comparison with RIS and skew-fit algorithms. We added some new experiments comparing with RIS and SkewFit algorithm in Appendix B.3 and B.5. We respond to the points brought by the reviewer below:
>
> **Q: Why $n=2$ for the negative binomial distribution? Is it assuming at least 2 steps?**
>
> **A:** The $n$ for negative binomial distribution corresponds to the planned path length. For example, $n = 2$ corresponds to finding a single waypoint that divides the task into two subtasks: start $\to$ waypoint and waypoint $\to$ goal. Our experiments use $n = 2$. Note that $n$ is not the number of time steps to reach the goal. The maximum steps of each episode can be found in Table 3 in Appendix C.2.
>
> **Q: The criticism about geometric distribution is not adequate.**
>
> **A:** We thank the reviewer for the advice. We would like to clarify that, we didn’t mean to criticize the geometric distribution. Indeed, it is a natural choice when training temporary difference algorithms and inspires our choice of prior. We just want to point out that it is also reasonable to consider other distributions (like negative binomial) other than geometric. We have updated the sentence in Paragraph 3 of Section 3.
>
> **Q: $s_g$ should pass to algorithm 2 C-Planning.**
>
> **A:** We thank the reviewer for pointing this out. We have updated the paper accordingly.
>
> **Q: Why are the experiments not the same as RIS?**
>
> **A:** We actually didn’t learn about RIS until we have already made up the environments. Another reason is that we want to explore practical applications of the algorithm as well as justify its effectiveness. Thus, we focused on the sequential robotic manipulation tasks in MetaWorld environments. We have also added the comparison of some very simple environments from the Ant-Maze tasks in Appendix B.5. On the simpler environments Ant-Small, we found that C-Planning performed slightly worse than RIS. In the harder environment Ant-Mid, we found RIS fails to reach the goal while C-Planning is consistently making progress.
>
> **Q: why not compare C-Planning to the skew-fit algorithm?**
>
> **A:** We implemented a version of the skew-fit algorithm on top of our framework and added an ablation study in Fig. 11 Appendix B.3 (brown line). We observe that our version of the Skew-Fit algorithm works well in relatively easy environments like Reach/Push, but fails on harder environments like PushWall.

---

> > ### Author Response · Authors · 2021-11-24
> > **Have the revisions addressed all the reviewer's concerns?**
> >
> > Dear Reviewer,
> >
> > Thank you for raising a number of excellent points in the review. Revising the paper to address these issues has made the paper more clear and precise. We would really appreciate a reply as to whether our response and clarifications have addressed the issues raised in the review, or whether there is anything else we can address.

---

### Official Review · Reviewer_gSmz · 2021-11-02

**Correctness:** 3
**Technical Novelty And Significance:** 3
**Empirical Novelty And Significance:** 3
**Recommendation:** 6
**Confidence:** 4

**Main Review:**

Strength:

1. The method nicely combines the C-learning with SoRB-like search methods. If my understanding is correct, we can regard this approach as a single-step waypoint search based on the learned universal value functions. Such an approach benefits from the fruitful goal-conditioned value function learning and can make progress in sparse-reward settings. At the same time, waypoint selection allows the agent to do structural exploration as suggested by "Why Does Hierarchy (Sometimes) Work So Well in Reinforcement Learning?" and get a better value estimation by expanding the expectation of values. According to my understanding, it is still hard to know if or why only searching waypoints during exploration should be better than searching the whole path (like in SoRB) and if we need to use the sampled waypoints to help us accelerate the bellman-equation update. However, I think the idea is already enough to serve as a strong competitor for goal-reaching tasks.
2. The method provides theoretical insights by modeling waypoint selection as an EM algorithm. At the same time, it shows good experiment results are good. I agree that the Obstacle-Drawer-Close task is pretty challenging. Success in such an environment is appealing.

Weakness:

1. My major concern is about the evaluation and the lack of link to the hierarchical RL methods. Many hierarchical RL methods exploit the way of combining a subgoal proposal network with a goal-condition function as C-Planning does. For example, the HIRO and HAC algorithms. In fact, we can also view LEAP, SoRB as kind of hierarchical RL algorithms in which high-level policies are model-based search methods. C-Planning employs a replay-buffer search based on the learned value function to propose subgoals. Though it is much simpler than previous approaches, the authors should at least provide a discussion about the relationship and the differences. I would also suggest authors conduct experiments on some classical environments such as AntMaze and AntPush to help researchers to position this work in literature.

Minor:

1. In Alg 1. line 7, there should be small bugs. There is no reason to concatenate the trajectory with itself.
2. I am a little confused by the performance of SoRB. Its performance decrease when it runs longer in Spiral 11x11. What's the reason behind it? I also do not understand why it can not solve this task. The simple maze environment should be very friendly to the graph-based search method.
3. The word CURRICULUM in the title is confusing. Do I miss any part that shows the proposed way-points are curriculum? I think curriculum refers to a process from the simple one to the hard one, but I don't understand why C-planning is a curriculum.

**Summary Of The Paper:**

This paper proposes C-Planning, a method that generates subgoals for distant goal-reaching tasks. The method is built upon the C-learning method, which learns a goal-condition policy together with a function that can estimate the connectivity between two states. The method then leverages an EM-like algorithm to propose waypoints by selecting a subgoal from replay buffer according to the distribution between the sum of state-to-waypoint distance and waypoint-to-goal distance. The authors apply this approach to both maze domain and challenging meta-world tasks and show improved performance.

**Summary Of The Review:**

This paper proposes a nice framework to combine goal-conditioned RL and waypoint selection. The method is neat and effective. Though the connection with the HRL is unclear from the paper, the advantages outweigh the flaws. I tend to accept the paper.

---

> ### Author Response · Authors · 2021-11-18
> **Author Response**
>
> We thank the reviewer for their positive review and thoughtful feedback. The reviewer was mainly concerned with how our work is positioned in the field of HRL and suggested some experiments in the Ant-Maze environments. We clarified the first point in our revised paper (Section 2) and ran some experiments in simple Ant-Maze tasks (Appendix B.5). We respond to the points brought by the reviewer below:
>
> **Q: The author should at least provide a discussion about the relationship and the differences between C-Planning and hierarchical RL methods.**
>
> **A:** We have updated the paper with a paragraph of the discussion on how C-Planning compares to hierarchical RL methods in paragraph 4 Section 2. As the reviewer also pointed out, C-Planning can be viewed as a special form of HRL where the high-level policy is a model-based search. However, traditional HRL methods often train a policy to predict subgoals [1, 2, 3]. C-Planning doesn’t train an additional policy for predicting subgoals. Instead, it is developed under a variational inference framework, adopts sampling from the replay buffer, and measures the distance between states by a classifier on the future state occupancy. Compared with approaches like LEAP [4] and SoRB [5], the waypoints sampling if C-Planning only happens at training time, saves a lot of computation time at deployment.
>
> **Q: I would also suggest authors conduct experiments on some classical environments such as AntMaze and AntPush to help researchers to position this work in literature.**
>
> **A**: As suggested by the reviewer, we ran an additional experiment using AntMaze, comparing C-Planning to RIS. The results of the experiments can be found in Appendix B.5. In these simple environments we designed, we only changed the maximum time steps from 600 to 300 (compared to RIS) and reset the agent to a fixed area instead of randomly at any position in the map. The details can be found in Appendix B. 5. On the simpler environments Ant-Small, we found that C-Planning performed slightly worse than RIS. In the harder environment Ant-Mid, we found RIS fails to reach the goal while C-Planning is consistently making progress.
>
> C-Planning was unable to solve the more complex environments (Ant-Maze $\pi$-Shape, Ant-Maze S-Shape). We attribute this failure to the fact that C-Planning doesn’t require or utilize any reward information or termination information from the environment while RIS uses both. This might also be due to the lack of hyperparameter tuning given a short time.
>
> **Q: Small bug in Alg.1 Line 7.**
>
> **A:** We thank the reviewers for pointing this out. The line 7 in Alg. 1 is in fact two part of a trajectory, $\tau_1 \sim p^\pi(\tau | s_0, s_w)$ and $\tau_2 \sim p^\pi(\tau | s_w, s_g)$. We have fixed this small typo and change them to $\tau_1 \sim p^\pi(\tau | s_0, s_w)$ and $\tau_2 \sim p^\pi(\tau | s_w, s_g)$.
>
> **Q: Performance of SoRB: “Its performance decreases when it runs longer in Spiral 11x11.**
>
> **A:** We think that the performance of SoRB at Spiral 11x11 is not getting worse when training longer. We would like to clarify that the SoRB performance is evaluated periodically and the starting point is randomly initialized. We took the checkpoints of C-Learning during training and performed SoRB on top of it. To be specific, we evaluate the performance of SoRB every 1e5 steps. Due to this reason, the performance curve has fewer sampling points and thus the variance might be a little higher. In addition, since the position of the agent is randomly initialized, it is also possible that during some evaluations the performance gets worse than before due to this type of variance. Due to these two reasons, we think the curve of the last point in SoRB at Spiral 11x11 is not necessarily indicating that the performance is worse but could just be variance.
>
> **Q: Performance of SoRB: “Why it can not solve this task of Spiral 11x11?**
>
> **A:** Our implementation of the environment is completely without any reward and termination function. In addition to that, as shown in Fig.3 (b), we always reset the starting position of the agent at the red point and command the goal at the green point (many goal-conditioned RL methods randomly reset the agent [4, 6]). Thus, we think that training in this way, the agent has very less signal about how to reach the goal and this may cause the SoRB to fail.
>
> **Q: The word “curriculum” is confusing in the title.**
>
> **A:** We use "curriculum" to refer to the sampling of the waypoint: the agent is initially tasked with reaching a subgoal, and the location of that subgoal changes throughout the course of training, resembling a curriculum. We have added a sentence to clarify this in Paragraph 4 in Section 1.

---

> > ### Author Response · Authors · 2021-11-18
> > **Author Response**
> >
> > **References:**
> >
> > [1] Gupta, Abhishek, et al. "Relay policy learning: Solving long-horizon tasks via imitation and reinforcement learning." arXiv preprint arXiv:1910.11956 (2019).
> >
> > [2] Levy, Andrew, et al. "Learning multi-level hierarchies with hindsight." arXiv preprint arXiv:1712.00948 (2017).
> >
> > [3] Nachum, Ofir, et al. "Data-efficient hierarchical reinforcement learning." arXiv preprint arXiv:1805.08296 (2018).
> >
> > [4] Nasiriany, ​​Soroush, et al. "Planning with goal-conditioned policies." arXiv preprint arXiv:1911.08453 (2019).
> >
> > [5] Eysenbach, Benjamin, Ruslan Salakhutdinov, and Sergey Levine. "Search on the replay buffer: Bridging planning and reinforcement learning." arXiv preprint arXiv:1906.05253 (2019).
> >
> > [6] Chane-Sane, Elliot, Cordelia Schmid, and Ivan Laptev. "Goal-conditioned reinforcement learning with imagined subgoals." International Conference on Machine Learning. PMLR, 2021.

---

> > ### Author Response · Authors · 2021-11-24
> > **Have the revisions addressed all the reviewer's concerns?**
> >
> > Dear Reviewer,
> >
> > We hope that you've had a chance to read our response. We would really appreciate a reply as to whether our response and clarifications have addressed the issues raised in the review, or whether there is anything else we can address.

---

> > > ### Comment · Reviewer_gSmz · 2021-11-30
> > > **Response to Authors**
> > >
> > > Thanks for your response. It clearly addressed my concerns. I will keep my score and recommend for acceptance.

---

### Decision · Program_Chairs · 2022-01-20

**Decision:**

Accept (Poster)

**Comment:**

All reviewers appreciate the suggested EM approach to goal-conditioned long-horizon reinforcement learning, and the technical contributions of the paper. While there is a mix in ratings, even the most critical reviewers feels that the paper has clear merits and is acceptable, and there are two solid acceptance recommendations. Overall, the papers significantly meets the standards of an ICLR paper acceptance.